# BAD inactivation exacerbates rheumatoid arthritis pathology by promoting survival of sublining macrophages

Jie Li[1,2,3†], Liansheng Zhang[1,3,4†], Yongwei Zheng[5], Rui Shao[1], Qianqian Liang[6], Weida Yu[1,2], Hongyan Wang[1], Weiguo Zou[1], Demin Wang[5], Jialing Xiang[7], Anning Lin[3,4]*

[1]The State Key Laboratory of Cell Biology, CAS Center for Excellence in Molecular Cell Science, Shanghai Institute of Biochemistry and Cell Biology, Chinese Academy of Sciences; University of Chinese Academy of Sciences, Shanghai, China; [2]School of Life Science and Technology, ShanghaiTech University, Shanghai, China; [3]Ben May Department for Cancer Research, The University of Chicago, Chicago, United States; [4]Institute of Modern Biology, Nanjing University, Nanjing, China; [5]Blood Research Institute, Blood Center of Wisconsin, Milwaukee, United States; [6]Department of Orthopaedics, Longhua Hospital, Shanghai University of Traditional Chinese Medicine, Shanghai, China; [7]Department of Biology, Illinois Institute of Technology, Chicago, United States

**Abstract** The resistance of synovial sublining macrophages to apoptosis has a crucial role in joint inflammation and destruction in rheumatoid arthritis (RA). However, the underlying mechanism is incompletely understood. Here we report that inactivation of the pro-apoptotic BCL-2 family protein BAD is essential for survival of synovial sublining macrophage in RA. Genetic disruption of *Bad* leads to more severe joint inflammation and cartilage and bone damage with reduced apoptosis of synovial sublining macrophages in collagen-induced arthritis (CIA) and TNFα transgenic (TNF-Tg) mouse models. Conversely, *Bad*[3SA/3SA] mice, in which BAD can no longer be inactivated by phosphorylation, are protected from collagen-induced arthritis. Mechanistically, phosphorylation-mediated inactivation of BAD specifically protects synovial sublining macrophages from apoptosis in highly inflammatory environment of arthritic joints in CIA and TNF-Tg mice, and in patients with RA, thereby contributing to RA pathology. Our findings put forward a model in which inactivation of BAD confers the apoptosis resistance on synovial sublining macrophages, thereby contributing to the development of arthritis, suggesting that BAD may be a potential therapeutic target for RA.

**\*For correspondence:**
anninglin@nju.edu.cn

**†**These authors contributed equally to this work

**Competing interests:** The authors declare that no competing interests exist.

## Introduction

Rheumatoid arthritis (RA) is an inflammatory autoimmune disease that primarily affects diarthrodial joints, characterized by hyperplasia of intimal lining, infiltration of macrophages, and lymphocytes in synovial sublining and joint destruction (*Firestein, 2003*; *McInnes and Schett, 2011*). Macrophage is one of the most abundant cell types in RA synovium and different subgroups of synovial macrophages play distinct roles in the development of RA (*Udalova et al., 2016*). Locally self-renewing tissue-resident macrophages, which mainly locate in synovial lining layer, have anti-inflammatory function and maintain tissue homeostasis for intra-articular structures (*Ambarus et al., 2012*; *Culemann et al., 2019*; *Uderhardt et al., 2019*). By contrast, the infiltrating macrophages derived from monocytes, which largely locate in synovial sublining layer in inflamed synovium, are the major

source of joint inflammation and cause cartilage and bone destruction by secreting pro-inflammatory cytokines including TNFα, IL-1β and IL-6, and matrix metalloproteinases (MMPs) (*Kennedy et al., 2011*; *Siouti and Andreakos, 2019*). Reduced number of synovial sublining macrophages correlates with clinical improvement in RA patients (*Bresnihan et al., 2009*; *Haringman et al., 2005*) and elimination of macrophages by clodronate-containing liposomes prevents arthritis development in murine models and patients with RA (*Barrera et al., 2000*; *Richards et al., 1999*; *Solomon et al., 2005*), indicating a critical role of the synovial sublining macrophages in promoting RA progression.

Defective apoptosis contributes to the survival and accumulation of synovial sublining macrophages (*Li et al., 2012*; *Pope, 2002*), which is an early RA hallmark that correlates with joint inflammation and bone destruction (*Tak et al., 1997*; *Udalova et al., 2016*). In inflamed joint, persistent activation of a variety of signaling pathways in macrophages, such as NF-κB signaling pathway and PI3K-Akt signaling pathway (*Handel et al., 1995*; *Liu et al., 2001*; *Vergadi et al., 2017*), protects macrophages from cell death under inflammatory condition in RA synovium. In addition, upregulated expressions of several anti-apoptotic proteins such as myeloid cell leukemia 1 (MCL-1) (*Liu et al., 2006*), cellular FLICE (FADD-like IL-1β-converting enzyme)-inhibitory protein (c-FLIP) (*Huang et al., 2017*), transcription factor nuclear factor of activated T cells 5 (NFAT5) (*Choi et al., 2017*), and microRNA such as *Mir-155* (*Kurowska-Stolarska et al., 2011*; *Rajasekhar et al., 2017*) have also been reported to promote synovial macrophage survival via inhibiting apoptosis in RA. However, direct genetic evidence that is responsible for the apoptosis resistance of synovial sublining macrophages is still lacking. Although genetic disruption of pro-apoptotic protein BID or BIM aggravates arthritis in K/BxN mouse model (*Scatizzi et al., 2006*; *Scatizzi et al., 2007*) and BH3 domain mimetic peptide treatment has also been reported to ameliorate arthritis development (*Scatizzi et al., 2010*), the underlying mechanism and targeting cell type are not known.

The pro-apoptotic BCL-2 family protein BAD has a critical role in mitochondrial-dependent apoptosis and involves in the development of many diseases by regulating cell death, such as diabetes, tumorigenesis, epilepsy, and sepsis shock (*Danial et al., 2008*; *Foley et al., 2018*; *Sastry et al., 2014*; *Yan et al., 2018*). The pro-apoptotic activity of BAD is inhibited by survival factors or growth factors such as IL-3 and EGF through activation of a group of protein kinases, such as Rsk2, PKA, Akt/PKB, and JNK1, which phosphorylate BAD at Ser112, Ser136, Ser155, and Thr201, respectively (*Danial, 2008*; *Yu et al., 2004*). Upon withdrawing survival factors or growth factors, dephosphorylated BAD will translocate to mitochondria, where it inactivates pro-survival BCL-2 family proteins BCL-2 and BCL-X$_L$ to trigger apoptosis (*Yang et al., 1995*). The pro-apoptotic activity of BAD is also inhibited by TNFα through activation of inhibitor of nuclear factor kappa-B kinase (IKK), which phosphorylates BAD at Ser26. The Ser26-phosphorylation primes BAD to be further phosphorylated at Ser112, Ser136, and Ser155 by other protein kinases, thereby preventing BAD from translocation to mitochondria to induce apoptosis (*Yan et al., 2013*; *Yan et al., 2018*). In the chronic inflammatory environment of RA synovium, TNFα-induced IKK activation may inhibit BAD-dependent apoptosis in the sublining macrophage, while increased level of growth factors such as vascular endothelial growth factor (VEGF) and platelet-derived growth factor (PDGF), which are majorly produced by synovial macrophages and involved in angiogenesis, fibrosis, and synovial inflammation (*Szekanecz and Koch, 2007*), may also be responsible for BAD inactivation in macrophage by activating Akt (*Son et al., 2014*), thereby contributing to synovial sublining macrophage survival and RA progression. However, the role of BAD in the development of RA has yet to be studied. Here we report that phosphorylation-mediated inactivation of BAD is increased in synovial sublining macrophage both in arthritic joints of mice and RA patients and *Bad* loss aggravates arthritis in both CIA model and TNFα transgenic (TNF-Tg) mouse model by promoting survival of synovial sublining macrophages. Conversely, constitutive activation of BAD protects mice from collagen-induced arthritis (CIA). Our study demonstrates that inactivation of BAD has a critical role in promoting the progression of RA by regulating synovial sublining macrophage survival.

## Results

### *Bad* loss aggravates CIA

To investigate the role of BAD in RA, we established CIA model in *Bad$^{-/-}$* mice and wild-type (WT) littermates, since the murine CIA model resembles most pathological features of human RA

(*Schinnerling et al., 2019*). The disease incidence was much higher and the arthritis was much severer in *Bad*$^{-/-}$ mice compared with WT littermates in CIA model (*Figure 1A*). Histopathological analysis showed significantly increased synovitis, pannus formation, as well as cartilage and bone destruction in *Bad*$^{-/-}$ mice compared with WT littermates (*Figure 1B,C*). Although there was no detectable difference in bone phenotype between *Bad*$^{-/-}$ mice and WT littermates at basal level (*Figure 1B,D*), X-ray and Micro-CT revealed that bone destruction was severer in *Bad*$^{-/-}$ mice in CIA (*Figure 1D*). Consistently, *Bad*$^{-/-}$ mice had increased number of osteoclasts and higher mRNA expression levels of osteoclast marker *TRAP* and *Ctsk* in the joints compared with WT littermates in CIA model (*Figure 1—figure supplement 1A,B*). However, when osteoclast differentiation was induced from bone marrow-derived macrophages (BMDMs) isolated from WT and *Bad*$^{-/-}$ mice in vitro, there was no significant difference in the number or mRNA expression levels of *TRAP* and *Ctsk* between WT and *Bad*$^{-/-}$ osteoclasts (*Figure 1—figure supplement 1C,D*). These results suggest that overly activation of osteoclasts in the synovium of *Bad*$^{-/-}$ mice in CIA model is not likely the result of *Bad* loss-induced direct differentiation of osteoclasts, but most likely due to increased levels of pro-inflammatory cytokines such as TNFα, which is known to promote osteoclastogenesis (*Wei et al., 2005*; *Wu et al., 2017*). In support of this notion, the levels of circulating collagen II-specific IgG and its subtypes in the serum after immunization were significantly elevated in *Bad*$^{-/-}$ mice, as analyzed by ELISA (*Figure 1E*). Quantitative PCR (qPCR) revealed that the mRNA levels of pro-inflammatory cytokines such as TNFα, IL-6, IL-1β, and MMPs like MMP-3 and MMP-13 in the joint were significantly increased, while anti-inflammatory cytokine IL-10 was decreased in *Bad*$^{-/-}$ mice (*Figure 1F*). Consistently, ELISA analysis showed that the protein levels of pro-inflammatory cytokines such as TNFα, IL-6, and IL-1β were also increased in the serum of *Bad*$^{-/-}$ mice (*Figure 1G*). These results demonstrate that *Bad*$^{-/-}$ mice are more susceptible to CIA.

## *Bad* loss leads to accumulation of macrophages and B cells but not CD4$^+$ T cells in CIA

We wanted to know whether *Bad* loss affects macrophages, which have a crucial role in inflammation in CIA, and the infiltration of lymphocytes in synovium, as B cell-mediated humoral immunity and T cell-mediated cellular immunity are known to play important roles in the pathogenesis of RA (*McInnes and Schett, 2011*). Flow cytometry analysis showed accumulation of macrophages was significantly increased in the synovium of *Bad*$^{-/-}$ mice compared with WT littermates in CIA model (*Figure 2—figure supplement 1A,B*). The percentage of synovial B cells was also increased in *Bad*$^{-/-}$ mice with CIA (*Figure 2—figure supplement 1C,D*), consistent with upregulated collagen II-specific antibodies production in the serum of *Bad*$^{-/-}$ mice in CIA model (*Figure 1E*). However, the percentage of synovial CD4$^+$ T cells was comparable between *Bad*$^{-/-}$ mice and WT littermates (*Figure 2—figure supplement 1E,F*). Consistently, *Bad* loss did not affect expression levels of CD4$^+$ T cell-associated cytokines including IFN-γ, IL-4, and IL-17 between WT and *Bad*$^{-/-}$ mice with CIA, although increased expression of IL-21 that is known to promote B cell differentiation and proliferation (*Dienz et al., 2009*; *Liu and King, 2013*; *Zotos et al., 2010*; *Figure 2—figure supplement 1G*). By contrast, the percentages of both CD4$^+$ T cells and B cells in spleen and lymph nodes (LNs) had no significant differences (*Figure 2—figure supplement 2*). These results suggest that *Bad* loss results in accumulation of macrophages and B cells but not CD4$^+$ T cells in synovium in mice with CIA.

## *Bad*-deficient B cells alone are not sufficient to promote the development of CIA

To determine the role of *Bad*-deficient B cells in the development of CIA, bone marrow cells isolated from *Bad*$^{-/-}$ mice and WT littermates were mixed with the bone marrow cells isolated from mature B cell-deficient μMT mice at a ratio of 1:4, respectively, and then transferred into lethally irradiated μMT mice to establish CIA model (*Figure 2—figure supplement 3A*). The percentage of B cells in peripheral blood was comparable between μMT mice transferred with WT (μMT:WT mice) and *Bad*$^{-/-}$ (μMT:*Bad*$^{-/-}$ mice) bone marrow cells after transplantation (*Figure 2—figure supplement 3B*). The incidence and severity of CIA were similar between μMT:WT mice and μMT:*Bad*$^{-/-}$ mice (*Figure 2—figure supplement 3C,D*). In support of this notion, histopathological analysis showed that there was no significant difference in synovitis, pannus formation, as well as bone and cartilage destruction between these two different groups of mice (*Figure 2—figure supplement 3E,F*).

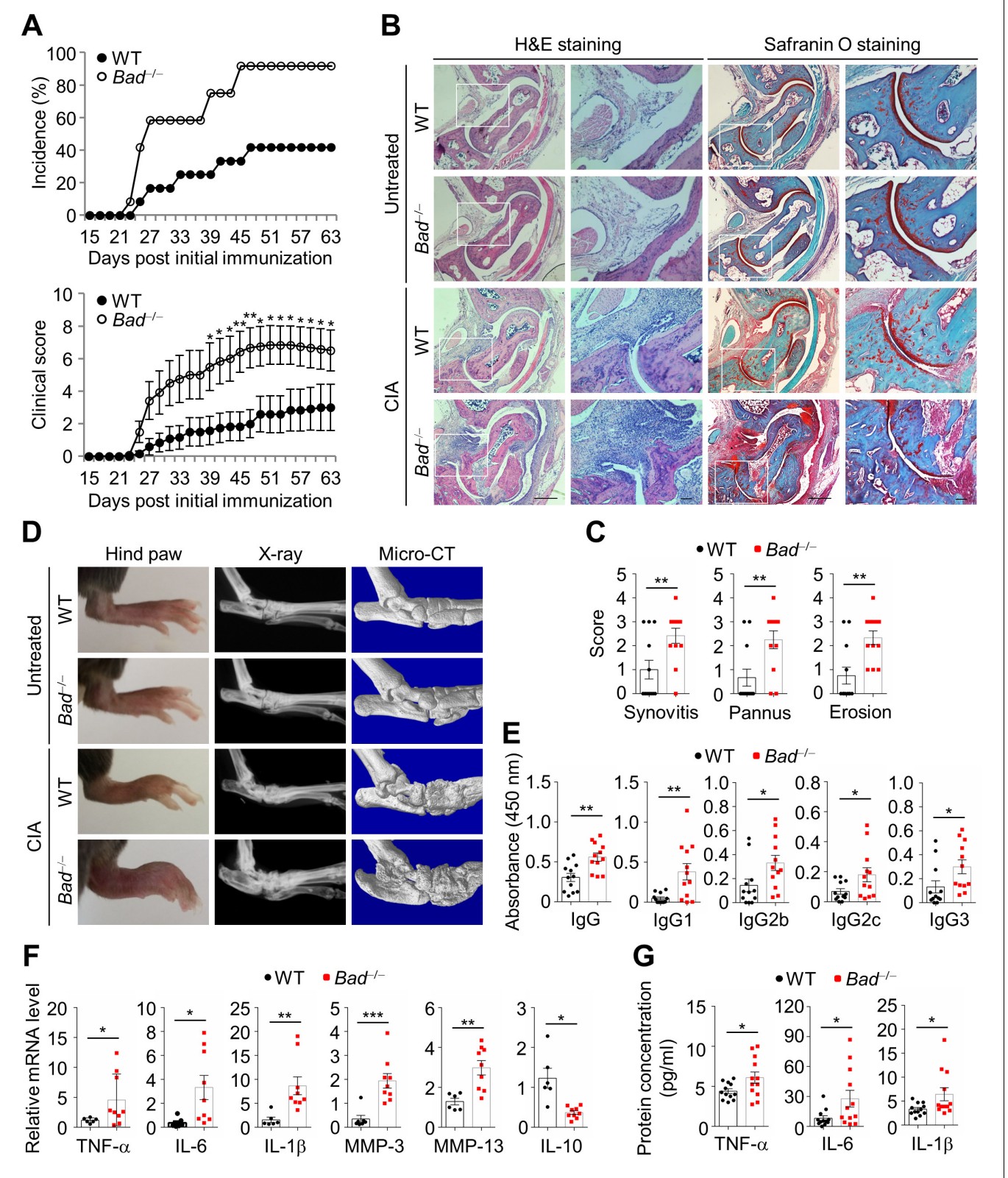

**Figure 1.** *Bad* loss aggravates collagen-induced arthritis (CIA). (**A**) Incidence and clinical scores of wild-type (WT; n = 12) and *Bad*−/− (n = 12) mice in CIA model. (**B**) H & E and Safranin O staining in ankle joint sections of WT and *Bad*−/− mice in CIA model and untreated control mice, ankle joints were harvested 63 days after primary immunization. Scale bar, 500 µm; magnified inset, 100 µm. (**C**) Evaluation of synovitis, pannus, and erosion of ankle joints of WT (n = 12) and *Bad*−/− (n = 12) mice in CIA model. (**D**) Representative photographs and radiographs of the hind paws of WT and *Bad*−/−

*Figure 1 continued on next page*

*Figure 1 continued*

mice in CIA model and untreated control mice. (E) Circulating levels of collagen II-specific antibodies in serum of WT (n = 12) and *Bad*$^{-/-}$ (n = 12) mice in CIA model were determined by ELISA. (F) Relative mRNA expression levels of pro-inflammatory cytokines (TNFα, IL-6, and IL-1β), matrix degradation enzymes (MMP-3 and MMP-13), and anti-inflammatory cytokine IL-10 in the joints of WT (n = 6) and *Bad*$^{-/-}$ (n = 9) mice in CIA model were determined by real-time qPCR. (G) Protein levels of pro-inflammatory cytokines TNFα, IL-6, and IL-1β in the serum of WT (n = 12) and *Bad*$^{-/-}$ (n = 12) mice in CIA model were determined by ELISA. All data are presented as mean ± SEM, and dots represent individual mice. Significant difference was analyzed by Mann–Whitney *U*-test (A) or unpaired Student's *t*-test (C, E, F, and G), *p<0.05; **p<0.01; ***p<0.001.

The online version of this article includes the following source data and figure supplement(s) for figure 1:

**Source data 1.** Source data for graphs in *Figure 1A,C,E,F, and G*.
**Figure supplement 1 .** BAD deficiency does not affect osteoclast activation in vitro.
**Figure supplement 1—source data 1.** Source data for graphs in *Figure 1—figure supplement 1*.

Consistently, *Bad* loss did not affect the autoantibody production, as measured by ELISA that detects anti-dsDNA IgG level in serum of 5-month-old WT and *Bad*$^{-/-}$ mice (*Figure 2—figure supplement 3G*). Thus, *Bad*-deficient B cells alone are not sufficient to promote the development of CIA. Most likely, the function of B cells in *Bad*$^{-/-}$ mice with CIA is augmented extrinsically, thereby contributing to CIA pathogenesis.

## *Bad* loss protects synovial macrophages from apoptosis in CIA

We hypothesized that *Bad* loss reduces apoptosis of synovial cells in mice with CIA. Immunoblotting analysis of cleaved Caspase-3 (cCasp-3), one of the executioner caspases in BAD-mediated apoptosis pathway, in the joint extracts of WT and *Bad*$^{-/-}$ mice with CIA revealed that the Casp-3 activity was decreased in *Bad*$^{-/-}$ mice (*Figure 2—figure supplement 4A*). Immunofluorescence double staining of TUNEL with anti-Vimentin (marker of fibroblast), anti-CD4 (marker of CD4$^+$ T cell), or anti-CD45R (marker of B cell) revealed that there was no significant difference in cell death in fibroblasts, CD4$^+$ T cells, or B cells in the synovium of *Bad*$^{-/-}$ and WT littermate mice in CIA model (*Figure 2—figure supplement 4B–D*). By contrast, death of macrophage as detected by TUNEL and anti-F4/80 double staining was significantly reduced in the synovium of *Bad*$^{-/-}$ mice (*Figure 2—figure supplement 4E*). Consistently, immunofluorescence double staining of anti-F4/80 and anti-cleaved Casp-3 revealed that macrophage apoptosis in sublining was significantly reduced, which was most likely responsible for accumulation of macrophages in the synovium of *Bad*$^{-/-}$ mice with CIA (*Figure 2A,B*). Consistently, flow cytometry analysis of cleaved caspase-3 positive synovial macrophages isolated from arthritic joints of WT and *Bad*$^{-/-}$ mice in CIA model showed reduced ratio of apoptotic macrophages in the joints of *Bad*$^{-/-}$ mice compared with WT mice in CIA model (*Figure 2C,D*). In addition, flow cytometry analysis of Annexin V showed reduced ratio of apoptotic macrophages in the joints of *Bad*$^{-/-}$ mice in CIA model (*Figure 2—figure supplement 5*). By contrast, there was no significant difference in the ratio of apoptotic B cells, CD4$^+$ T cells, or synovial fibroblasts in arthritic joints of WT and *Bad*$^{-/-}$ mice in CIA model, analyzed by either cleaved caspase-3 (*Figure 2—figure supplement 6*) or Annexin V (*Figure 2—figure supplement 7*) flow cytometry assay. These results indicate that *Bad* loss mainly affects macrophage apoptosis, while apoptosis of other cell types appear to be less affected. Suppression of macrophage apoptosis may not only promote inflammation but also further stimulate B cell proliferation, thereby aggravating CIA in *Bad*$^{-/-}$ mice.

Next we wanted to know whether BAD involves in TNFα-induced macrophage apoptosis, as RA is a chronic inflammatory disease which has consistent expression of TNFα, while inactivation of BAD by IKK can inhibit TNFα-induced apoptosis (*Yan et al., 2013*). When WT and *Bad*$^{-/-}$ BMDMs were pretreated with IKK-specific inhibitor PS1145 to block TNFα-induced IKK activation, *Bad*$^{-/-}$ BMDMs were resistant to TNFα-induced apoptosis in vitro, as analyzed by flow cytometry (*Figure 2E,F*), indicating *Bad* loss can protect synovial macrophages from TNFα-induced apoptosis, thereby augmenting CIA.

## *Bad*-deficient macrophages sufficiently aggravate CIA

To determine the role of *Bad* loss in macrophages in the development of CIA, bone marrow cells isolated from WT or *Bad*$^{-/-}$ mice (both on CD45.2 background) were transferred into lethally irradiated

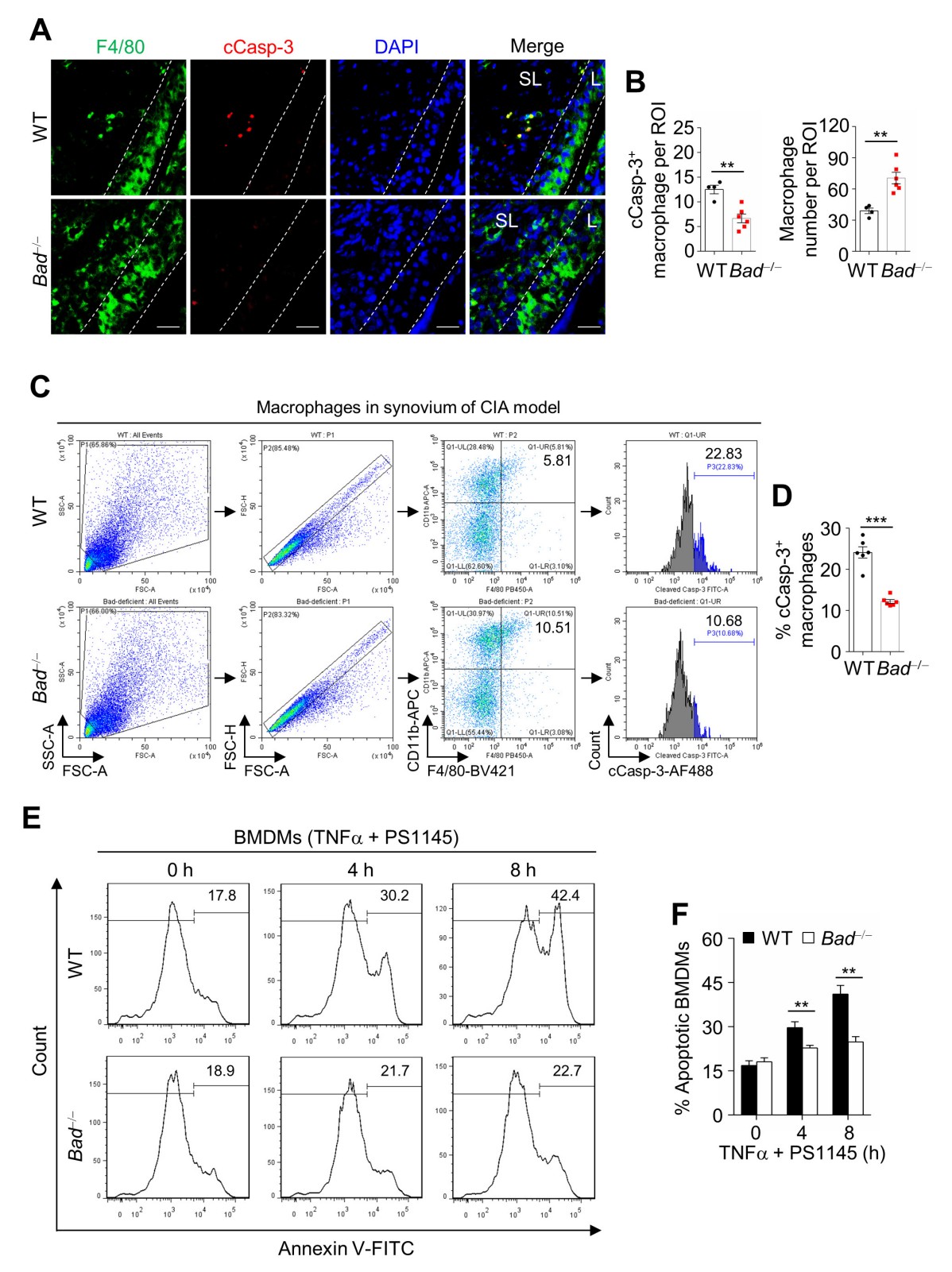

**Figure 2.** *Bad* loss protects synovial macrophages from apoptosis in collagen-induced arthritis (CIA). (**A**) Double staining of anti-F4/80 and anti-cleaved Casp-3 in the ankle joint sections of wild-type (WT) and *Bad*$^{-/-}$ mice in CIA model. L: lining; SL: Sublining. Scale bar, 25 μm. (**B**) Quantification of the number of cleaved Casp-3 positive macrophage and total macrophage per region of interest (ROI) in the ankle joint sections of WT (n = 4) and *Bad*$^{-/-}$ (n = 6) mice in CIA model. (**C**) Apoptotic synovial macrophages from WT and *Bad*$^{-/-}$ mice in CIA model were analyzed by flow cytometry by staining

*Figure 2 continued on next page*

*Figure 2 continued*

cleaved Casp-3. (**D**) Quantification of the ratio of cleaved Casp-3 positive synovial macrophages in WT (n = 6) and $Bad^{-/-}$ (n = 6) mice in CIA model. (**E**) Apoptotic cell death of WT and $Bad^{-/-}$ bone marrow-derived macrophages (BMDMs) pretreated with PS1145 (10 μM) for 2 hr, followed by stimulation with TNFα (5 ng/ml) for various times as indicated, was analyzed by flow cytometry. (**F**) Quantification of apoptotic BMDMs for various time points as indicated. Data in (**E and F**) represent two individual experiments with similar results. All data are presented as mean ± SEM; dots represent individual mice. Significant difference was analyzed by unpaired Student's $t$-test, \*\*p<0.01; \*\*\*p<0.001.

The online version of this article includes the following source data and figure supplement(s) for figure 2:

**Source data 1.** Source data for graphs in *Figure 2B,D, and F*.
**Source data 2.** Source data for graphs in *Figure 2—figure supplements 1–7*.
**Figure supplement 1.** The percentages of macrophage, B cell, and CD4+ T cell in synovium of wild-type (WT) and $Bad^{-/-}$ mice in collagen-induced arthritis (CIA) model.
**Figure supplement 2.** The number of B cell and CD4+ T cell in spleen and lymph node (LN) has no significant difference between wild-type (WT) and $Bad^{-/-}$ mice in collagen-induced arthritis (CIA) model.
**Figure supplement 3.** *Bad*-deficient B cells alone are not sufficient to promote collagen-induced arthritis (CIA).
**Figure supplement 4.** *Bad* loss inhibits apoptosis in macrophages but not in fibroblasts, CD4+ T cells, or B cells in collagen-induced arthritis (CIA).
**Figure supplement 5.** *Bad* loss protects synovial macrophages from apoptosis in collagen-induced arthritis (CIA).
**Figure supplement 6.** *Bad* loss does not significantly affect the apoptosis of B cells, CD4+ T cells, or synovial fibroblasts in collagen-induced arthritis (CIA) detected by cleaved Casp-3 staining.
**Figure supplement 7.** *Bad* loss does not significantly affect the apoptosis of B cells, CD4+ T cells, or synovial fibroblasts in collagen-induced arthritis (CIA) detected by Annexin V staining.

WT recipient mice (CD45.1 background), which had a complete and stable chimerism with donor mice in bone marrow after transplantation (*Figure 3—figure supplement 1*). The recipient mice were subjected to CIA 6 weeks after transplantation (*Figure 3A*), among which the mice transferred with $Bad^{-/-}$ bone marrow cells (CD45.1:$Bad^{-/-}$ mice) displayed more severe arthritis and higher incidence and clinical score than the mice transferred with WT bone marrow cells (CD45.1:WT mice) (*Figure 3B,C*). Histopathological analysis showed increased synovitis, pannus formation, as well as cartilage and bone destruction in CD45.1:$Bad^{-/-}$ mice compared with CD45.1:WT mice (*Figure 3D, E*). The number of synovial sublining macrophages in CD45.1:$Bad^{-/-}$ mice was also increased compared with that in CD45.1:WT mice, while apoptosis of synovial sublining macrophages in CD45.1:$Bad^{-/-}$ mice was significantly reduced, as analyzed by double staining of anti-F4/80 and anti-cleaved Casp-3 (*Figure 3F,G*). Thus, *Bad*-deficient macrophages are sufficient to promote the development of CIA, suggesting that macrophage function is augmented intrinsically in $Bad^{-/-}$ mice to contribute to CIA pathology.

## The pro-apoptotic activity of BAD was suppressed in synovial sublining macrophages in CIA

We next determined how the pro-apoptotic activity of BAD is regulated in synovial sublining macrophages in mice with CIA. There was no significant difference in BAD protein level in total joint extracts between control (non-arthritic) and arthritic joints of WT mice in CIA (*Figure 4A*). By contrast, phosphorylation of BAD at the 'regulatory serines' (Ser112, Ser136, and Ser155), indicative of BAD inactivation (*Datta et al., 2002*) was profoundly increased in arthritic joints compared with control joints (*Figure 4A*). Immunohistochemistry (IHC) staining revealed that Ser136-phosphorylated BAD [pBAD(S136)] was significantly increased mainly in the synovial sublining layer of arthritic joint compared with that in the control joint (*Figure 4B,C*). Furthermore, the cells with strong pBAD (S136) signal were mostly localized in the cartilage-pannus junction areas (*Figure 4B*), which are often infiltrated by macrophages (*Tak and Bresnihan, 2000*). In support of this notion, immunofluorescence double staining of anti-pBAD(S136) along with different cell markers revealed that pBAD (S136) mainly located in macrophages (F4/80+), but not T cells (CD3+), B cells (CD45R+) or fibroblasts (Vimentin+) (*Figure 4D,E*). This was not the result of cell-type specific expression of BAD (*Figure 4—figure supplement 1*), and little pBAD(S136) signal was found in these cell types in non-arthritic joints (*Figure 4—figure supplement 2*). More importantly, pBAD(S136)+ macrophages mainly located in the sublining layer of arthritic joint synovium (*Figure 4F*). These data indicate that inactivation of BAD is responsible for, at least in part, the resistance of infiltrated macrophages to apoptosis in CIA.

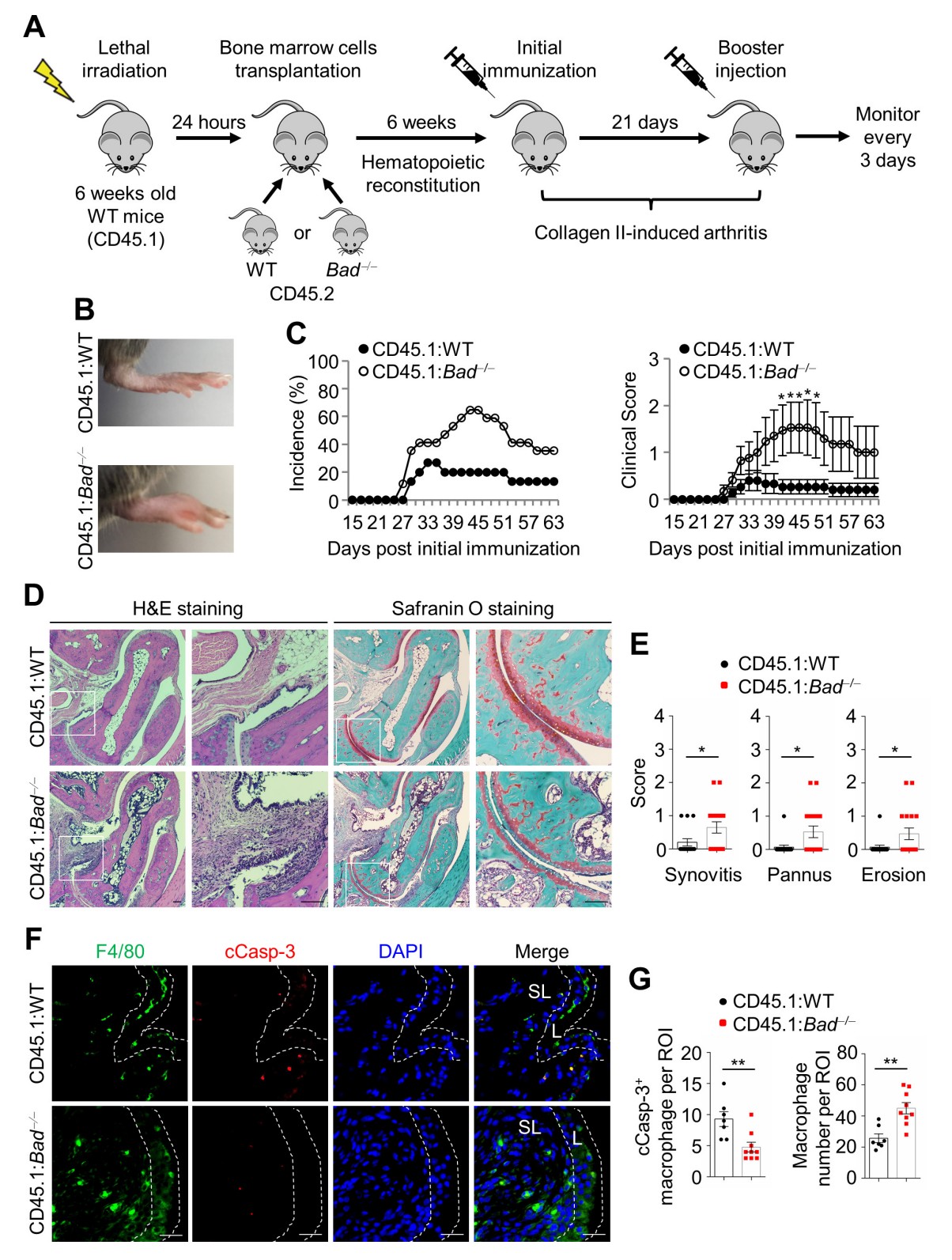

**Figure 3.** *Bad*-deficient macrophages sufficiently aggravate collagen-induced arthritis (CIA). (**A**) The experimental design diagram for bone marrow transplantation in CD45.1 background mice followed by CIA model. (**B**) Representative photographs of the hind paws from CD45.1:WT and CD45.1: *Bad*[−/−] mice in CIA model. (**C**) Incidence and clinical scores of CD45.1:WT (n = 15) and CD45.1:*Bad*[−/−] (n = 17) mice in CIA model. (**D**) H & E and Safranin O staining in ankle joint sections of CD45.1:WT and CD45.1:*Bad*[−/−] mice in CIA model, ankle joints were harvested 63 days after primary

*Figure 3 continued on next page*

*Figure 3 continued*

immunization. Scale bar, 100 µm. (E) Evaluation of synovitis, pannus, and erosion of ankle joints from CD45.1:WT (n = 15) and CD45.1:*Bad*$^{-/-}$ (n = 17) mice in CIA model. (F) Double staining of anti-F4/80 and anti-cleaved Casp-3 in the ankle joint sections of CD45.1:WT and CD45.1:*Bad*$^{-/-}$ mice in CIA model. Scale bar, 25 µm. (G) Quantification of the number of cleaved Casp-3 positive macrophage and total macrophage per field in the ankle joint sections of CD45.1:WT (n = 7) and CD45.1:*Bad*$^{-/-}$ (n = 9) mice in CIA model. All data are presented as mean ± SEM; dots represent individual mice. Significant difference was analyzed by Mann–Whitney *U*-test (C) or unpaired Student's *t*-test (E and G), *p<0.05; **p<0.01.

The online version of this article includes the following source data and figure supplement(s) for figure 3:

**Source data 1.** Source data for graphs in *Figure 3C,E, and G*.
**Figure supplement 1.** Efficiency of bone marrow transplantation in CD45.1 mice.
**Figure supplement 1—source data 1.** Source data for graphs in *Figure 3—figure supplement 1*.

To elucidate the signaling pathways that lead to inactivation of BAD in synovial macrophages in CIA, we examined the activity of Akt and IKK in the synovial macrophages in the control and arthritic joints of WT mice with CIA, since Akt is a known BAD(S136) kinase upon stimulation by growth factors and survival factors and is upregulated in the synovium of RA for survival of synovial cells (*Datta et al., 1997*; *Jiao et al., 2018*), while IKK is known to phosphorylate BAD at Ser26, which is a prerequisite for BAD to be phosphorylated at Ser112, Ser136, and Ser155 upon stimulation by TNFα (*Yan et al., 2013*), which is the key pro-inflammatory cytokine that induces inflammation in RA (*van Schouwenburg et al., 2013*). Immunofluorescence double staining of anti-pAkt and anti-F4/80 revealed that macrophages in the sublining synovium of arthritic joints had significantly increased activation of Akt compared with that in control joints (*Figure 5A,C*), so was activation of IKK (*Figure 5B,D*). Consistently, phosphorylation of Akt and IKK, as well as the protein levels of PDGF and VEGF were significantly increased in the arthritic joint extracts (*Figure 5E*). Taken together, activation of Akt and IKK are responsible for, at least in part, phosphorylation and inactivation of BAD in infiltrated sublining macrophages in CIA model.

## Inactivation of BAD in synovial sublining macrophages in TNF-Tg mice

We also determined the pathological function of BAD in TNF-Tg mice, another murine model of experimental arthritis in which overexpression of human TNFα leads to inflammatory-erosive arthritis (*Li and Schwarz, 2003*). Histopathological analysis showed increased synovitis, pannus formation, as well as cartilage and bone destruction in the joints of 3-month-old TNF-Tg/*Bad*$^{-/-}$ mice compared with TNF-Tg littermates (*Figure 6A,B*), suggesting that *Bad* loss also exacerbates experimental arthritis in TNF-Tg model. IHC staining revealed that pBAD(S136) was significantly increased in the synovial sublining layer of TNF-Tg mice compared with that in the control mice (*Figure 6C,D*) and was co-stained with anti-F4/80 (*Figure 6E*), suggesting that pBAD(S136)$^+$ macrophages were mainly increased in the synovial sublining layer. More importantly, the macrophage apoptosis, as revealed by anti-F4/80 and anti-cleaved Casp-3 double staining, was significantly reduced in the synovial sublining layer in TNF-Tg/*Bad*$^{-/-}$ mice compared with the control TNF-Tg littermates, correlating with increased macrophage numbers in synovial sublining layer in TNF-Tg/*Bad*$^{-/-}$ mice (*Figure 6—figure supplement 1*). Consistently, flow cytometry analysis of cleaved caspase-3 also revealed reduced macrophage apoptosis under the same conditions (*Figure 6F,G*). These results suggest that BAD phosphorylation and inactivation is a shared mechanism for apoptosis resistance of synovial sublining macrophage in two different experimental arthritis murine models.

## *Bad*$^{3SA/3SA}$ mice are resistant to CIA

To demonstrate phosphorylation-mediated inactivation of BAD contributes to the development of experimental arthritis, we performed CIA in *Bad*$^{3SA/3SA}$ knockin mice, in which the regulatory serine sites were mutated to alanines so that the resultant BAD mutant can no longer be phosphorylated and inactivated (*Datta et al., 2002*; *Figure 7A,B*). *Bad*$^{3SA/3SA}$ mice displayed very lower incidence and significantly reduced severity of arthritis compared with WT littermates in CIA model (*Figure 7C,D*). Histopathological analysis showed significantly decreased synovitis, pannus formation, as well as cartilage and bone destruction in *Bad*$^{3SA/3SA}$ mice (*Figure 7E,F*).

We next determined whether the decreased incidence and reduced severity of CIA in *Bad*$^{3SA/3SA}$ mice is the result of increased apoptosis of synovial sublining macrophages. Immunoblotting analysis

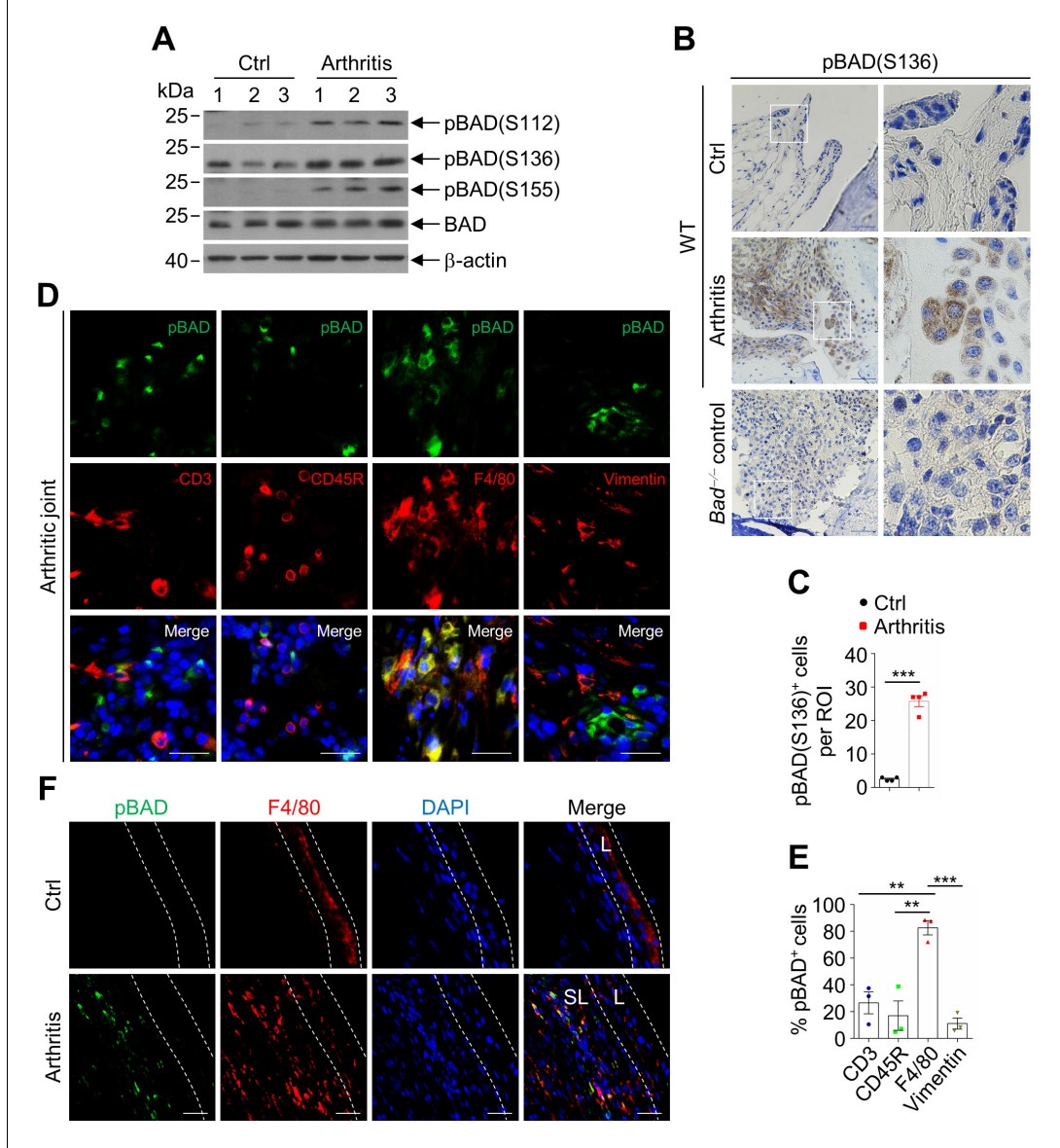

**Figure 4.** The pro-apoptotic activity of BAD was suppressed in synovial sublining macrophages in collagen-induced arthritis (CIA). (**A**) Immunoblotting analysis of pBAD(S112), pBAD(S136), pBAD(S155), and BAD in ankle joints extracts of wild-type (WT) control (non-arthritic) and arthritic mice in CIA model (n = 3). (**B**) Immunohistochemistry staining of anti-pBAD(S136) in the synovium of WT control and arthritic mice in CIA model, *Bad*^−/− mice serve as control. Scale bar, 50 μm. (**C**) Quantification of the number of pBAD(S136) positive cells per field (n = 4). (**D**) Double staining of anti-pBAD(S136) with T cell marker anti-CD3, B cell marker anti-CD45R, macrophage marker anti-F4/80, or synovial fibroblast marker anti-Vimentin respectively in the synovium of WT arthritic mice in CIA model. Scale bar, 25 μm. (**E**) Quantification of percentages of pBAD(S136) positive cells in total cells per field of different cell types (n = 3). (**F**) Double staining of anti-pBAD(S136) and anti-F4/80 in the synovium of WT control and arthritic mice in CIA model. Scale bar, 25 μm. All data are presented as mean ± SEM; dots represent individual mice. Significant difference was analyzed by unpaired Student's *t*-test (**C**) or one-way ANOVA test (**E**), **p<0.01; ***p<0.001.

The online version of this article includes the following source data and figure supplement(s) for figure 4:

**Source data 1.** Source data for graphs in *Figure 4C and E*.
**Source data 2.** Source data for graphs in *Figure 4—figure supplements 1* and *2*.
**Figure supplement 1.** The expression of BAD is not cell-type specific in arthritic joints in collagen-induced arthritis (CIA) model.
**Figure supplement 2.** pBAD staining in various cell types of non-arthritic joints.

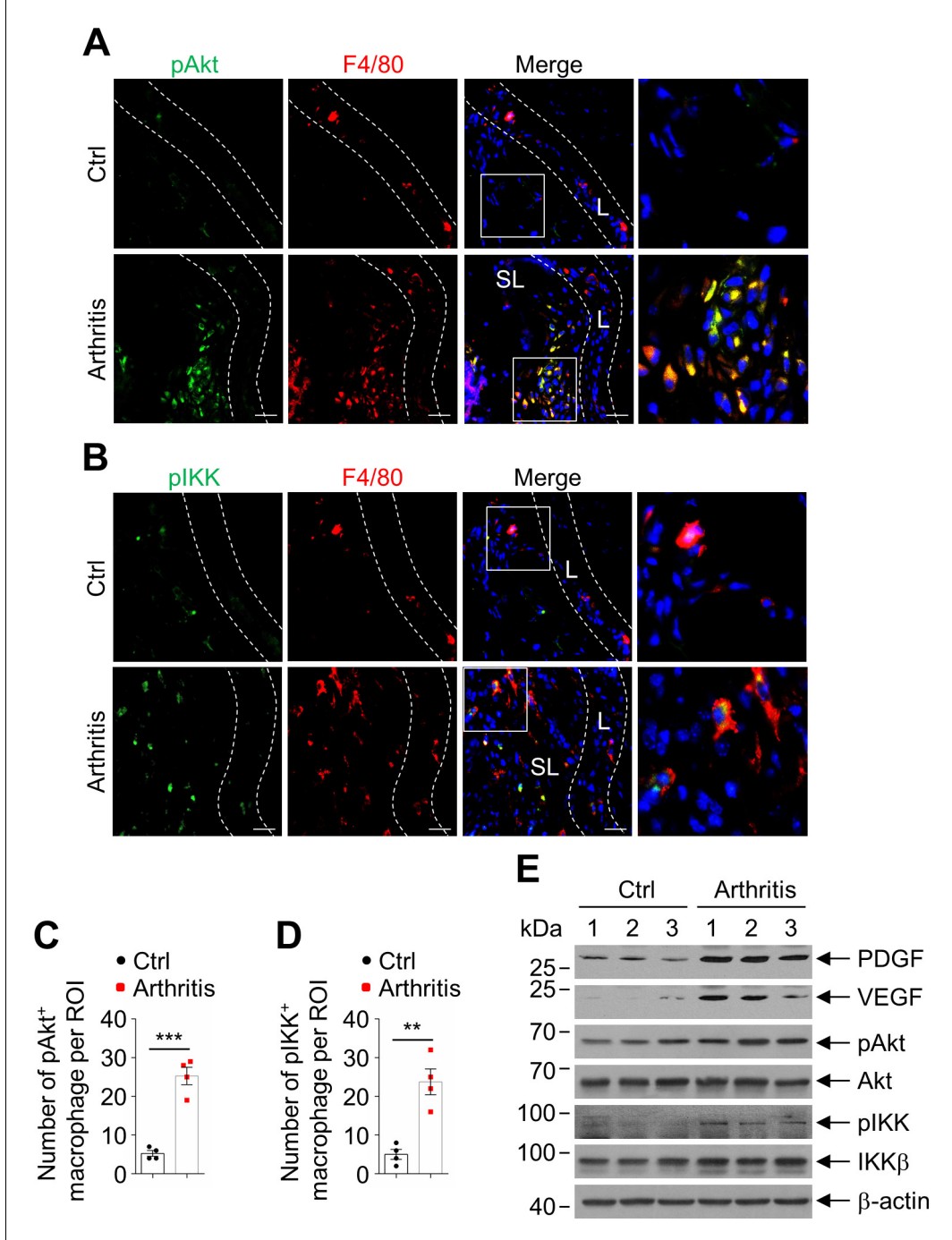

**Figure 5.** Akt and IKK activation was increased in synovial sublining macrophage in collagen-induced arthritis (CIA). (A) Double staining of anti-pAkt and anti-F4/80 in the synovium of wild-type (WT) control and arthritic mice in CIA model. Scale bar, 25 μm. (B) Double staining of anti-pIKK and anti-F4/80 in the synovium of WT control and arthritic mice in CIA model. Scale bar, 25 μm. (C) Quantification of the number of pAkt and F4/80 double positive cells per field in the synovium of WT control (n = 4) and arthritic (n = 4) mice in CIA model. (D) Quantification of the number of pIKK and F4/80 double positive cells per field in the synovium of WT control (n = 4) and arthritic (n = 4) mice in CIA model. (E) Immunoblotting analysis of the protein levels of platelet-derived growth factor (PDGF), vascular endothelial growth factor (VEGF), pAkt, Akt, pIKK, and IKK in the joint extracts of WT non-arthritic (n = 3) and arthritic (n = 3) mice in CIA model. All data are presented as mean ± SEM; dots represent individual mice. Significant difference was analyzed by unpaired Student's *t*-test, **p<0.01; ***p<0.001. The online version of this article includes the following source data for figure 5:

**Source data 1.** Source data for graphs in *Figure 5C and D*.

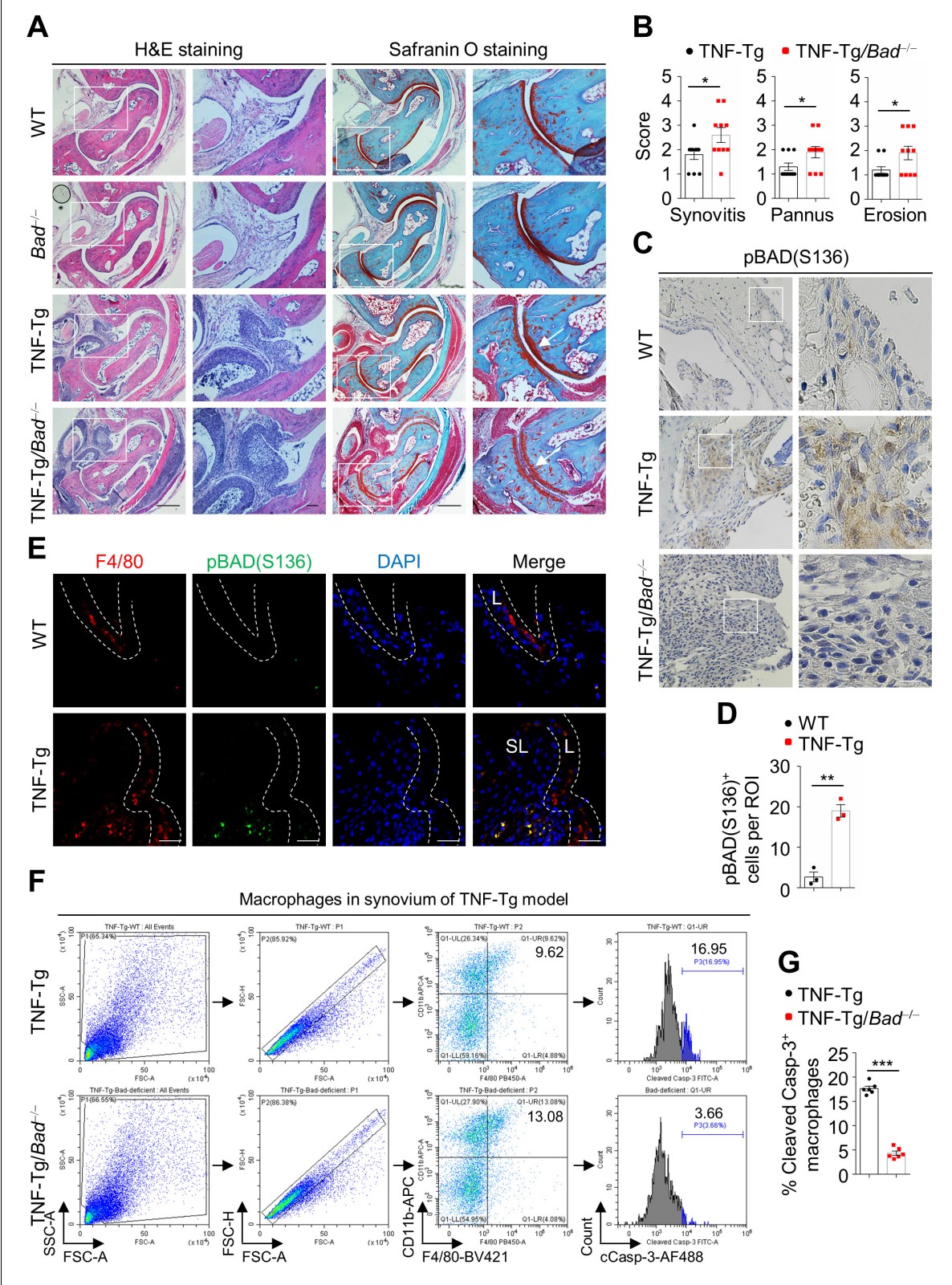

**Figure 6.** *Bad* loss aggravates arthritis in TNFα transgenic (TNF-Tg) mouse model. (**A**) H & E and Safranin O staining in ankle joint sections of 3-month-old wild-type (WT), *Bad*⁻/⁻, TNF-Tg, and TNF-Tg/*Bad*⁻/⁻ mice. Scale bar, 500 μm; magnified inset, 100 μm. (**B**) Evaluation of synovitis, pannus, and erosion of ankle joints of TNF-Tg (n = 10) and TNF-Tg/*Bad*⁻/⁻ (n = 10) mice. (**C**) Immunohistochemistry staining of anti-pBAD(S136) in the synovium of 3-month-old WT, TNF-Tg, and TNF-Tg/*Bad*⁻/⁻ mice. Scale bar, 50 μm. (**D**) Quantification of the number of pBAD(S136) positive cells per field (n = 3).
*Figure 6 continued on next page*

*Figure 6 continued*

(E) Double staining of anti-pBAD(S136) and anti-F4/80 in the synovium of 3-month-old WT control and TNF-Tg mice. Scale bar, 25 μm. (F) Apoptotic synovial macrophages from TNF-Tg and TNF-Tg/$Bad^{-/-}$ mice were analyzed by flow cytometry by staining cleaved Casp-3. (G) Quantification of the ratio of cleaved Casp-3 positive synovial macrophages in TNF-Tg (n = 6) and TNF-Tg/$Bad^{-/-}$ (n = 6) mice. All data are presented as mean ± SEM; dots represent individual mice. Significant difference was analyzed by unpaired Student's *t*-test, *p<0.05; **p<0.01; ***p<0.001.

The online version of this article includes the following source data and figure supplement(s) for figure 6:

**Source data 1.** Source data for graphs in *Figure 6B,D, and G*.

**Figure supplement 1.** *Bad* loss prevents sublining macrophages from apoptosis in TNFα transgenic (TNF-Tg) mouse model.

**Figure supplement 1—source data 1.** Source data for graphs in *Figure 6—figure supplement 1*.

showed increased protein level of cleaved Casp-3 in joint extracts of $Bad^{3SA/3SA}$ mice compared with WT littermates in CIA model (*Figure 7G*). Furthermore, immunofluorescence double staining of anti-F4/80 with anti-cleaved Casp-3 revealed that apoptosis of synovial sublining macrophage was significantly increased, consistent with decreased synovial macrophage number in $Bad^{3SA/3SA}$ mice compared with that in WT littermates (*Figure 7H,I*). Taken together, these results demonstrate that phosphorylation-mediated inactivation of BAD is critical for synovial sublining macrophages survival and the development of CIA.

## Inactivation of BAD in synovial sublining macrophages in patients with RA

We wondered whether BAD also plays an important role in human RA. H & E staining showed increased infiltration of inflammatory cells in the synovial sections of RA patients compared with osteoarthritis (OA) patients (*Figure 8A*), consistent with previous reports (*Kennedy et al., 1988*). IHC staining of anti-pBAD (S99, equivalent to S136 in mouse) revealed that phosphorylation of human BAD (hBAD) at Ser99 was significantly higher in sublining layer of synovium of RA patients compared with that in OA patients (*Figure 8A,B*), consistent with the results in mouse arthritic joints (*Figure 4B,C*). Immunofluorescence double staining of anti-pBAD(S99) along with the markers of several synovial cell types including macrophage (CD68⁺), B cell (CD20⁺), T cell (CD3⁺), and fibroblast (Vimentin⁺) revealed that pBAD(S99) was mainly located in macrophages in sublining but not B cells, T cells, or fibroblasts (*Figure 8C–E*), consistent with the observations in CIA mice (*Figure 4D–F*). This was not the result of cell-type-specific expression of BAD (*Figure 8—figure supplement 1*). These results demonstrate that phosphorylation of BAD was increased in macrophages in sublining synovium of RA patients, thereby contributing to the apoptotic resistance of pro-inflammatory macrophages and pathology in RA patients.

## Discussion

It has long been thought that the resistance of infiltrating synovial sublining macrophages to apoptosis plays a central role in RA pathogenesis (*Bresnihan et al., 2009*; *Haringman et al., 2005*; *Udalova et al., 2016*). Although several anti- and pro-apoptotic regulators have been reported to involve in regulation of apoptosis in activated macrophages, the causative relationship has yet to be established. In this report, we found that BAD phosphorylation, indicative of its inactivation, was increased in synovial sublining macrophages in arthritic mice and patients with RA. Using both loss-of-function and gain-of-function genetic approaches, we show that *Bad* loss augments the development of CIA by promoting survival and accumulation of synovial sublining macrophages, while constitutive activation of BAD prevents CIA progression by inducing apoptosis of synovial sublining macrophages. Thus, BAD is a key determinant for the development of experimental arthritis by regulating the survival of synovial sublining macrophages.

Our finding identifies BAD as a crucial player in experimental arthritis pathogenesis. Previous studies have shown that regulation of the maturation, activation, and function of different infiltrating immune cells in synovium can suppress or promote experimental arthritis pathogenesis in mice. For instance, genetic disruption of β-arrestin1 (*Li et al., 2013*), IFT20 (*Yuan et al., 2014*), and CD80/86 (*O'Neill et al., 2007*) suppresses arthritis development by impairing T cell differentiation and activation, while loss of NFAT5 (*Choi et al., 2017*), IKKβ (*Armaka et al., 2018*), CIKS (*Pisitkun et al., 2010*), and SIRT1 (*Woo et al., 2016*) inhibits arthritis pathogenesis by promoting apoptosis in

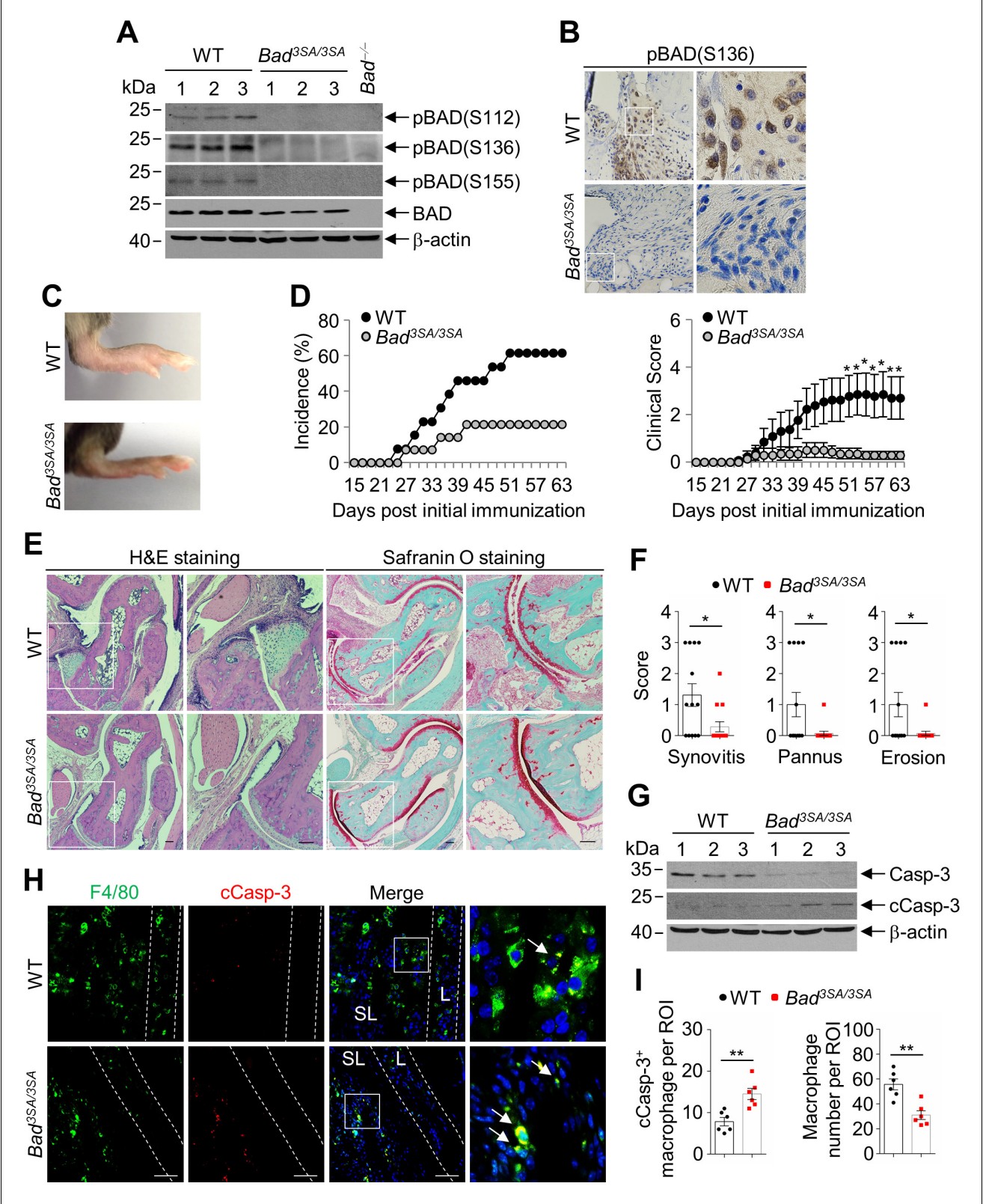

**Figure 7.** *Bad*^3SA/3SA mice are resistant to collagen-induced arthritis (CIA). (**A**) Immunoblotting analysis of pBAD(S112), pBAD(S136), pBAD(S155), and BAD in the joint extracts of wild-type (WT; n = 3) and *Bad*^3SA/3SA (n = 3) mice in CIA model. (**B**) Representative IHC staining of pBAD(S136) in the synovium of immunized WT and *Bad*^3SA/3SA mice. Scale bar, 50 μm. (**C**) Representative photographs of the hind paws of WT and *Bad*^3SA/3SA mice in CIA model. (**D**) Incidence and clinical scores of WT (n = 13) and *Bad*^3SA/3SA (n = 14) mice in CIA model. (**E**) H & E and Safranin O staining in ankle joint

*Figure 7 continued on next page*

*Figure 7 continued*

sections of WT and *Bad*^3SA/3SA mice in CIA model. Ankle joints were harvested 63 days after primary immunization. Scale bar, 100 μm. (**F**) Evaluation of synovitis, pannus, and erosion of ankle joints from WT (n = 13) and *Bad*^3SA/3SA (n = 14) mice in CIA model. (**G**) Immunoblotting analysis of caspase-3 and cleaved Casp-3 in ankle joints of immunized WT (n = 3) and *Bad*^3SA/3SA (n = 3) mice. (**H**) Double staining of anti-F4/80 and anti-cleaved Casp-3 in the sublining area of ankle joint sections of WT and *Bad*^3SA/3SA mice in CIA model. Scale bar, 50 μm. (**I**) Quantification of the cleaved Casp-3-positive macrophage and number of total macrophage per field in the ankle joint sections of WT (n = 6) and *Bad*^3SA/3SA (n = 6) mice in CIA model. All of the data are presented as mean ± SEM; dots represent individual mice. Significant difference was analyzed by Mann–Whitney *U*-test (**D**) or unpaired Student's *t*-test (**F and I**), *p<0.05; **p<0.01.

The online version of this article includes the following source data for figure 7:

**Source data 1.** Source data for graphs in *Figure 7D,F, and I*.

---

macrophages and fibroblasts, or inhibiting antibody production and dendritic cell maturation, respectively. On the other hand, genetic ablation of A20 (*Matmati et al., 2011*), PGRN (*Tang et al., 2011*), and CTRP6 (*Murayama et al., 2015*) increased the susceptibility of mice to experimental arthritis development by promoting NF-κB activation, TNFα-mediated inflammation and complement activation, respectively. Our results show that arthritis pathogenesis was exacerbated in *Bad*^−/− mice in both CIA and TNF-Tg murine models of RA, but protected in *Bad*^3SA/3SA mice in CIA model, due to reduced or augmented apoptosis in synovial sublining macrophages, thereby adding BAD to the list of the key players that determine the susceptibility of mice to experimental arthritis.

BAD differentially regulates the survival of various infiltrating immune cells in experimental arthritis. Previous studies have shown that infiltrating innate immune cells like macrophages and adaptive immune cells such as CD4^+ T cells and B cells in sublining synovium contributes to arthritis pathogenesis, as elimination of macrophages, CD4^+ T cells, or B cells prevents arthritis development (*Solomon et al., 2005*; *Svensson et al., 1998*; *Taneja et al., 2002*). Our results show that *Bad* loss led to accumulation of synovial sublining macrophages and B cells but not CD4^+ T cells, accompanying with reduced apoptosis in synovial sublining macrophages in CIA model. Consistently, *Bad* loss reduced apoptosis of synovial sublining macrophages in TNF-Tg mice, which is resulted from macrophage-mediated spontaneous inflammation with limited contribution of B cells and CD4^+ T cells (*Schinnerling et al., 2019*). These results indicate that synovial sublining macrophages were more dependent on inactivation of BAD to resist apoptosis than other immune cells in the arthritic joints. This is likely due to, at least in part, distinct expression levels of BAD in different types of infiltrating immune cells, as BAD expression level is high in monocytes, which are the major source of synovial sublining macrophages (*Udalova et al., 2016*), in bone marrow and peripheral blood but it is very low in spleen B cells and T cells and most tissue or organ fibroblasts (*Kitada et al., 1998*). Further studies using the mice with conditionally deleted or mutated *Bad* in macrophages are needed to unambiguously ascribe the role of BAD in macrophages, given the possibility that *Bad* loss might affect the radio sensitivity of hematopoietic cells during the transplantation. By contrast, *Bad* loss appears to affect accumulation of B cells extrinsically, as bone marrow transplantation of *Bad*-deficient B cells failed to affect CIA progression in μMT mice. It is possible that B cell accumulation is the result of increased survival of synovial sublining macrophages, which are known to stimulate B cell proliferation (*Craxton et al., 2003*). Although *Bad* loss did not affect CD4^+ T cell numbers in CIA model, expression of IL-21 that is mainly secreted by CD4^+ cells was significantly increased. Since IL-21 is known to promote B cell differentiation and proliferation (*Dienz et al., 2009*; *Liu and King, 2013*; *Zotos et al., 2010*), CD4^+ T cells may also contribute to B cell accumulation in *Bad*-deficient CIA mouse model. Future studies are needed to determine the mechanism by which *Bad* loss affects CD4^+ T cell functions. Detailed mechanism apart, inactivation of BAD reduces apoptosis of synovial sublining macrophages intrinsically but increases accumulation of B cells extrinsically in CIA.

BAD is inactivated by Akt and IKK in synovial sublining macrophages in CIA. Previous studies have shown that the pro-apoptotic activity of BAD is inactivated by a group of protein kinases including Rsk2, PKA, Akt/PKB, and JNK1, which phosphorylate BAD at Ser112, Ser136, Ser155, or Thr201 (*Danial, 2008*; *Yu et al., 2004*), and IKK (*Yan et al., 2013*; *Yan et al., 2018*), which phosphorylates BAD at Ser26 and primes its phosphorylation by other protein kinases, in response to survival factors/growth factors such as IL-3 and EGF or pro-inflammatory cytokine TNFα, respectively. Our results show that BAD in synovial sublining macrophages was inactivated by Akt and IKK in CIA model. Phosphorylation and activation of Akt and IKK were significantly increased in synovial

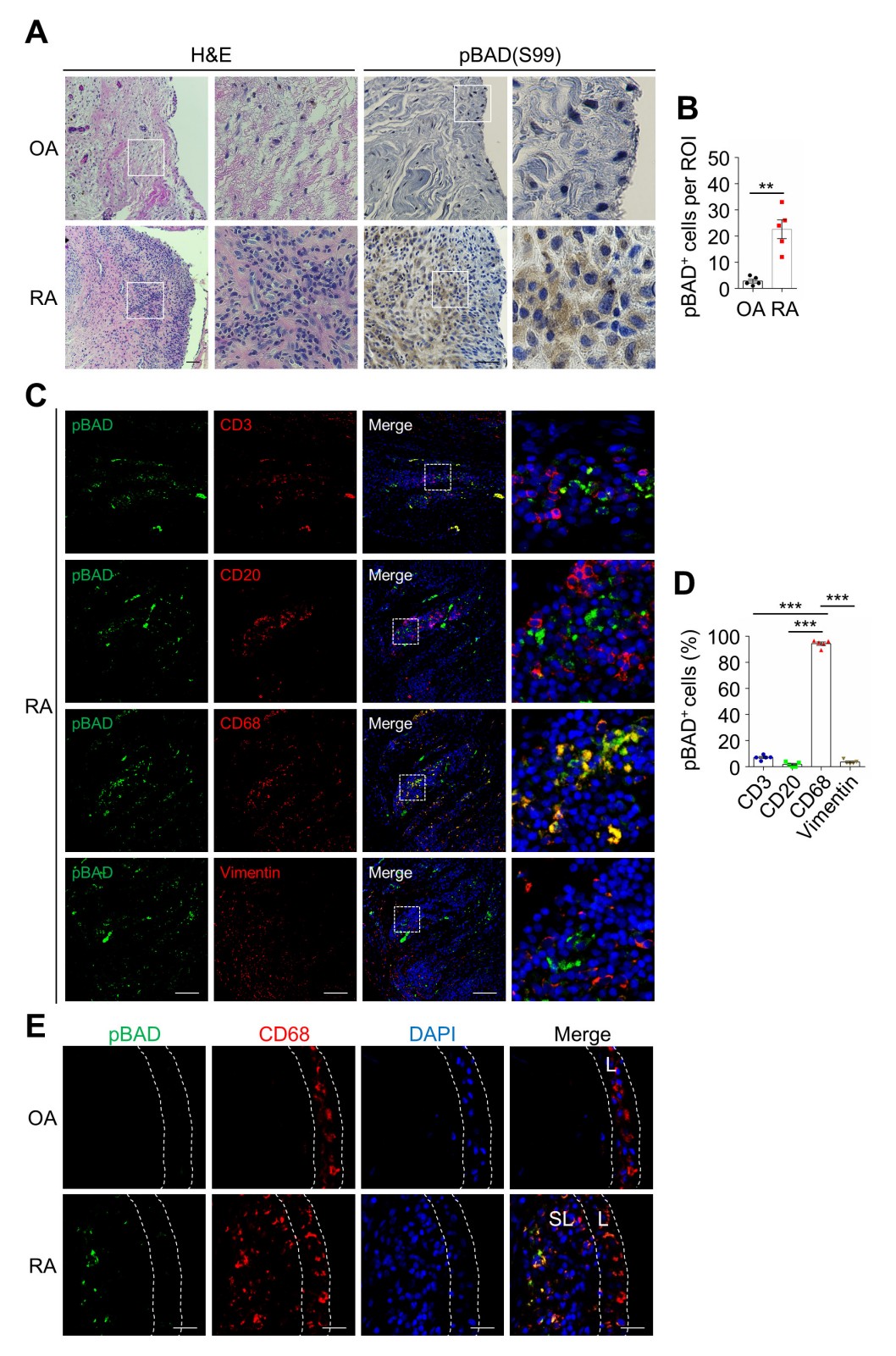

**Figure 8.** BAD phosphorylation is increased in the synovial sublining macrophages of rheumatoid arthritis (RA) patients. (**A**) H & E and pBAD(S99) immunohistochemistry staining of the synovium specimen sections of patients with OA and RA. Scale bar, 100 μm. (**B**) Quantification of pBAD(S99)-positive cells per field in the synovial specimen sections of patients with OA (n = 5) and RA (n = 5). (**C**) Double staining of anti-pBAD(S99) with T cell marker anti-CD3, B cell marker anti-CD20, macrophage marker anti-CD68 or synovial fibroblast marker anti-Vimentin in serial synovial specimen sections

*Figure 8 continued on next page*

*Figure 8 continued*

of patients with RA. Scale bar, 100 μm. (D) Quantification of the percentage of pBAD(S99)-positive cells in total cells per field of different cell types (n = 5). (E) Double staining of anti-pBAD(S99) and anti-F4/80 in the synovial specimen sections of patients with OA and RA. Scale bar, 25 μm. All data are presented as mean ± SEM; dots represent individual human samples. Significant difference was analyzed by unpaired Student's *t*-test (B) or one-way ANOVA test (D), **p<0.01, ***p<0.001.

The online version of this article includes the following source data and figure supplement(s) for figure 8:

**Source data 1.** Source data for graphs in *Figure 8B and D*.
**Figure supplement 1.** The expression of BAD is not cell-type specific in synovium of rheumatoid arthritis (RA) patients.
**Figure supplement 1—source data 1.** Source data for graphs in *Figure 8—figure supplement 1*.

sublining macrophages in the arthritic joints, in which increased levels of growth factors such as VEGF and PDGF are known to involve in inflammation, angiogenesis, and fibrosis (*Tas et al., 2016*), while TNFα is known to mediate inflammation, angiogenesis, and osteoclasts formation and activation (*McInnes et al., 2016*). Akt-mediated BAD Ser136-phosphorylation was also significantly increased in synovial sublining macrophages in CIA and TNF-Tg mice. Thus, BAD is the convergent node for Akt and IKK signaling pathways to confer the resistance to apoptosis on synovial sublining macrophages, thereby contributing to the pathogenesis of arthritis.

Our finding has important pathological significance and clinic relevance. The current RA therapies are mainly centered on anti-inflammation strategy aiming at reducing the detrimental effects of pro-inflammatory cytokines such as TNFα, IL-1, and IL-6 in RA, along with the utilization of immune-suppressors and small chemical JAK inhibitors (*Firestein and McInnes, 2017*). However, the therapeutic efficacy varies among the patients with RA and the risk of overly infection due to immune suppression increases significantly over the time. Our results show that inactivation of BAD protects synovial sublining macrophages from apoptosis, thereby contributing to experimental arthritis pathology in both CIA and TNF-Tg murine models, and more importantly BAD inactivation by phosphorylation was significantly increased in synovial sublining macrophages in human specimens. Thus, reinvigoration of BAD pro-apoptotic activity in synovial sublining macrophages could provide a potential specific therapeutic strategy for RA therapy.

# Materials and methods

**Key resources table**

| Reagent type (species) or resource | Designation | Source or reference | Identifiers | Additional information |
|---|---|---|---|---|
| Biological sample (*Homo sapiens*) | Synovial tissue of RA and OA patients | The University of Chicago | Department of Pathology's diagnostic archive | Paraffin-embedded slides |
| Strain, strain background *Mus musculus* | CD45.1 mice (C57BL/6 congenic) | The Jackson Laboratory | Stock #:002014 RRID:IMSR_JAX:002014 | B6.SJL-*Ptprc^a Pepc^b*/BoyJ |
| Strain, strain background *Mus musculus* | μMT mice (C57BL/6 congenic) | The Jackson Laboratory | Stock #:002288 RRID:IMSR_JAX:002288 | B6.129S2-*Ighm^{tm1Cgn}*/J |
| Antibody | Anti-BAD (rabbit polyclonal) | Cell Signaling Technology | Cat #:9292 RRID:AB_331419 | Immunoblotting (1:500) |
| Antibody | Anti-pBAD(S136) (rabbit monoclonal) | Cell Signaling Technology | Cat #:4366 RRID:AB_10547878 | Immunoblotting (1:500) Immunofluorescence (1:100) Immunohistochemistry (1:100) |
| Antibody | Anti-pBAD(S112) (rabbit polyclonal) | Cell Signaling Technology | Cat #:9291 RRID:AB_331417 | Immunoblotting (1:500) |
| Antibody | Anti-pBAD(S155) (rabbit polyclonal) | Cell Signaling Technology | Cat #:9297 RRID:AB_2062131 | Immunoblotting (1:500) |
| Antibody | Anti-pAkt(S473) (rabbit monoclonal) | Cell Signaling Technology | Cat #:4060 RRID:AB_2315049 | Immunoblotting (1:1000) Immunofluorescence (1:100) |
| Antibody | Anti-Akt (rabbit monoclonal) | Cell Signaling Technology | Cat #:4691 RRID:AB_915783 | Immunoblotting (1:1000) |

*Continued on next page*

*Continued*

| Reagent type (species) or resource | Designation | Source or reference | Identifiers | Additional information |
|---|---|---|---|---|
| Antibody | Anti-pIKKα/β (rabbit monoclonal) | Cell Signaling Technology | Cat #:2078 RRID:AB_2079379 | Immunoblotting (1:500) Immunofluorescence (1:100) |
| Antibody | Anti-IKKβ (rabbit polyclonal) | Cell Signaling Technology | Cat #:2684 RRID:AB_2122298 | Immunoblotting (1:500) |
| Antibody | Anti-cleaved Caspase-3 (rabbit polyclonal) | Cell Signaling Technology | Cat #:9661 RRID:AB_2341188 | Immunoblotting (1:500) Immunofluorescence (1:100) |
| Antibody | Anti-Caspase-3 (rabbit polyclonal) | Cell Signaling Technology | Cat #:9662 RRID:AB_331439 | Immunoblotting (1:500) |
| Antibody | Alexa Fluor 488-conjugated cleaved Caspase-3 (rabbit polyclonal) | Cell Signaling Technology | Cat #:9669 RRID:AB_2069869 | Flow cytometry (1:100) |
| Antibody | Anti-Vimentin (chicken polyclonal) | Novus Biologicals | Cat #:NB300-223 RRID:AB_10003206 | Immunofluorescence (1:100) |
| Antibody | Anti-BAD (rabbit monoclonal) | Abcam | Cat #:ab32445 RRID:AB_725614 | Immunofluorescence (1:100) |
| Antibody | Anti-CD11b (rabbit monoclonal) | Abcam | Cat #:ab133357 RRID:AB_2650514 | Immunofluorescence (1:200) |
| Antibody | Anti-F4/80 (rat monoclonal) | Abcam | Cat #:ab6640 RRID:AB_1140040 | Immunofluorescence (1:200) |
| Antibody | Anti-CD45R (rat monoclonal) | Abcam | Cat #:ab64100 RRID:AB_1140036 | Immunofluorescence (1:200) |
| Antibody | Anti-CD3 (rat monoclonal) | Abcam | Cat #:ab11089 RRID:AB_369097 | Immunofluorescence (1:200) |
| Antibody | Anti-Vimentin (mouse monoclonal) | Abcam | Cat #:ab8069 RRID:AB_306239 | Immunofluorescence (1:200) |
| Antibody | Anti-CD68 (mouse monoclonal) | Dako | Cat #:M0876 RRID:AB_2074844 | Immunofluorescence (1:200) |
| Antibody | Anti-β-actin (mouse monoclonal) | Santa Cruz | Cat #:sc-47778 RRID:AB_2714189 | Immunoblotting (1:2000) |
| Antibody | APC-conjugated anti-mouse CD11b (rat monoclonal) | Biolegend | Cat #:101211 RRID:AB_312794 | Flow cytometry (1:100) |
| Antibody | BV421-conjugated anti-mouse F4/80 (rat monoclonal) | Biolegend | Cat #:123131 RRID:AB_10901171 | Flow cytometry (1:100) |
| Antibody | APC-conjugated anti-mouse CD4 (rat monoclonal) | Biolegend | Cat #:100411 RRID:AB_312696 | Flow cytometry (1:100) |
| Antibody | BV421-conjugated anti-mouse CD4 (rat monoclonal) | Biolegend | Cat #:100437 RRID:AB_10900241 | Flow cytometry (1:100) |
| Antibody | APC-conjugated anti-mouse/human CD45R/B220 (rat monoclonal) | Biolegend | Cat #:103211 RRID:AB_312996 | Flow cytometry (1:100) |
| Antibody | BV421-conjugated anti-mouse B220 (rat monoclonal) | Biolegend | Cat #:103239 RRID:AB_10933424 | Flow cytometry (1:100) |
| Antibody | BV421-conjugated anti-mouse CD45 (rat monoclonal) | Biolegend | Cat #:103133 RRID:AB_10899570 | Flow cytometry (1:100) |
| Antibody | FITC-conjugated anti-mouse CD45 (rat monoclonal) | Biolegend | Cat #:103107 RRID:AB_312972 | Flow cytometry (1:100) |

*Continued*

| Reagent type (species) or resource | Designation | Source or reference | Identifiers | Additional information |
|---|---|---|---|---|
| Antibody | APC-conjugated anti-mouse CD45.1 (mouse monoclonal) | Biolegend | Cat #:110713 RRID:AB_313502 | Flow cytometry (1:100) |
| Antibody | APCCy7-conjugated anti-mouse CD45.2 (mouse monoclonal) | Biolegend | Cat #:109824 RRID:AB_830789 | Flow cytometry (1:100) |
| Antibody | Alexa Fluor 594-conjugated anti-mouse Vimentin (mouse monoclonal) | Biolegend | Cat #:677804 RRID:AB_2566179 | Flow cytometry (1:100) |
| Antibody | PE-conjugated anti-mouse Gr1 (rat monoclonal) | eBioscience | Cat #:12-5931-82 RRID:AB_466045 | Flow cytometry (1:100) |
| Antibody | APC-conjugated anti-mouse Mac1 (rat monoclonal) | eBioscience | Cat #:17-0112-82 RRID:AB_469343 | Flow cytometry (1:100) |
| Commercial assay or kit | FITC Annexin V Apoptosis Detection Kit I | BD Pharmingen | Cat #:556547 | Flow cytometry (5 µl) |
| Commercial assay or kit | In Situ Cell Death Detection Kit | Roche | Cat #:11684817910 | TUNEL staining |
| Commercial assay or kit | Mouse IL-1β ELISA Kit | R and D Systems | Cat #:MLB00C | ELISA |
| Commercial assay or kit | Mouse IL-6 ELISA Kit | R and D Systems | Cat #:M6000B | ELISA |
| Commercial assay or kit | Mouse TNFα ELISA Kit | R and D Systems | Cat #:MTA00B | ELISA |
| Commercial assay or kit | Mouse anti-dsDNA IgG ELISA Kit | Alpha Diagnostic | Cat #:5120 | ELISA |
| Commercial assay or kit | Leukocyte Acid Phosphatase Kit | Sigma-Aldrich | Cat #:387A | TRAP staining |
| Software, algorithm | GraphPad Prism 6.0 | GraphPad Prism | RRID:SCR_002798 | http://www.graphpad.com/ |
| Software, algorithm | ImageJ | ImageJ | RRID:SCR_003070 | https://imagej.net/ |
| Software, algorithm | FlowJo 10 | FlowJo | RRID:SCR_008520 | https://www.flowjo.com/solutions/flowjo |
| Software, algorithm | CytExpert | CytExpert Software | RRID:SCR_017217 | https://www.beckman.fr/flow-cytometry/instruments/cytoflex/software |
| Other | Chicken type II collagen | Chondrex | Cat #:20012 | 2 mg/ml |
| Other | Complete Freund's adjuvant (CFA) | Chondrex | Cat #:7023 | 5 mg/ml heat-denatured mycobacterium |

## Patient samples

Paraffin-embedded slides of synovial tissue from RA patients and OA patients (five each) were obtained from the Department of Pathology's diagnostic archive of the University of Chicago with diagnostic reports with patients' consent. The mean age of RA patients was 56 years old, ranging from 33 to 67 years old (female/male = 1.5). The mean age of OA patients was 65.4 years old, ranging from 55 to 75 (female/male = 1.5). This study was reviewed and approved by an Institutional Review Board (IRB) at the University of Chicago.

## Mice

*Bad*$^{-/-}$ and *Bad*$^{3SA/3SA}$ mice have been previously described (**Datta et al., 2002**; **Ranger et al., 2003**). Briefly, *Bad*$^{-/-}$ and *Bad*$^{3SA/3SA}$ mice have been backcrossed into the C57BL/6J genetic background for at least 14 generations and validated by genome scanning to be 99.9% congenic with C57BL/6J. Heterozygous mice were further bred to generate knockout/knockin experimental mice

and WT littermates. CD45.1 (B6.SJL-$Ptprc^a$ $Pepc^b$/BoyJ, #002014) mice and μMT (B6.129S2-$Ighm^{tm1Cgn}$/J, #002288) mice were purchased from the Jackson Laboratory. The 3647 line of TNF-Tg mice was generated by Dr. George Kollias (Institute of Immunology, Alexander Fleming Biomedical Sciences Research Center, Vari, Greece) (*Keffer et al., 1991*). The TNF-Tg mice were bred as heterozygotes on a C57BL/6J background. TNF-Tg/$Bad^{-/-}$ mice were generated by crossbreeding $Bad^{-/-}$ mice with TNF-Tg mice. Only male mice were used and were randomly chosen for each genotype with age-matched. For clinical score of the mice and microscopic analysis of immunofluorescence staining, the experimenters were blinded to each genotype. All of the mice were maintained under specific pathogen-free conditions. Animal studies were approved by the Institutional Animal Care and Use Committee of Shanghai Institute of Biochemistry and Cell Biology or the Institutional Animal Care and Use Committee of the University of Chicago.

## Collagen-induced arthritis

CIA in C57BL/6 background mice was established, as previously reported (*Inglis et al., 2008*). Briefly, 2 mg/ml chicken type II collagen (20012, Chondrex) was emulsified with an equal volume of complete Freund's adjuvant (CFA) containing 5 mg/ml heat-denatured mycobacterium (7023, Chondrex). Eight to twelve weeks old mice were immunized intradermally at several sites near the base of the tail with 100 μl emulsion. A booster injection was administered on day 21 with the same emulsion of collagen II and CFA. The mice were monitored every other day after booster immunization. Development of arthritis was evaluated as described previously (*Campbell et al., 2000*).

## Bone marrow transplantation

For bone marrow transplantation in CD45.1 mice, 6-week-old male recipient mice (CD45.1 background) were injected with one million bone marrow cells from donor mice of $Bad^{-/-}$ or WT littermates (CD45.2 background) in 100 μl PBS within 24 hr after lethal irradiation of the recipient mice with two doses of 540 rad (total 1080 rad) delivered at least 2 hr apart by RS 2000 X-ray irradiator. The recipient mice were administrated Uniprim diet 1 week prior to irradiation and continuously treated with Uniprim diet for 2 weeks after transplantation to against infection. Six weeks later, the recipient mice were immunized with collagen II and CFA to establish the CIA model.

For bone marrow transplantation in μMT mice, the bone marrow cells from $Bad^{-/-}$ and WT littermates were mixed with bone marrow cells from μMT mice as the ratio of 1:4 and then injected into lethally irradiated μMT mice (6-week-old) with $5 \times 10^6$ cells in 100 μl PBS per recipient within 24 hr after lethal irradiation. Six weeks later, the mice were immunized with collagen II and CFA to establish the CIA model.

## Radiography

Control and immunized WT and $Bad^{-/-}$ mice were euthanized with $CO_2$ 63 days after the primary immunization and the hind paws were removed and fixed in 70% ethanol and analyzed by X-ray (Faxitron X-ray MX-20 Specimen Radiography System) and Micro-CT (Skyscan1172, Bruker Biospin) instrument.

## Histopathology

Hind limbs of the control mice or mice with CIA were removed, fixed in 4% PFA for 48 hr, decalcified in 15% EDTA (pH 7.8), and embedded in paraffin. Serial sections of ankle joints at 5 μm were cut and stained with H & E and Safranin O. The evaluation of synovitis, pannus formation, as well as bone and cartilage destruction were determined by a graded scale as described previously (*Tang et al., 2011*).

## Antibodies

Antibodies against BAD (#9292), pBAD(S136) (#4366), pBAD(S112) (#9291), pBAD(S155) (#9297), pAkt(S473) (#4060), Akt (#4691), pIKKα/β (#2078), IKKβ (#2684), cleaved Caspase-3 (#9661), Caspase-3 (#9662), and Alexa Fluor 488-conjugated cleaved Caspase-3 (#9669) were from Cell signaling Technologies (CST). Antibody against mouse Vimentin (NB300-223) was from Novus Biologicals. Antibodies against BAD (ab32445), CD11b (ab133357), F4/80 (ab6640), CD45R (ab64100), CD3 (ab11089), and human Vimentin (ab8069) were from Abcam. Antibodies against CD68 (M0876) and

CD20 (IS604) were from Dako. Antibody against β-actin (sc-47778) was from Santa Cruz. APC-conjugated anti-mouse CD11b (M1/70) (101211), BV421-conjugated anti-mouse F4/80 (BM8) (123131), FITC-conjugated anti-mouse F4/80 (BM8) (11-4801-81), APC-conjugated anti-mouse CD4 (GK1.5) (100411), Alexa Fluor 594-conjugated anti-mouse Vimentin (O91D3) (677804), APC-conjugated anti-mouse B220 (RA3-6B2) (50-0452-82), APC-conjugated anti-mouse/human CD45R/B220 (RA3-6B2) (103211), BV421-conjugated anti-mouse B220 (RA3-6B2) (103239), PE-conjugated anti-mouse B220 (RA3-6B2) (12-0452-83), FITC-conjugated anti-mouse CD4 (GK1.5) (11-0041-85), BV421-conjugated anti-mouse CD4 (GK1.5) (100437), BV421-conjugated anti-mouse CD45 (30-F11) (103133), FITC-conjugated anti-mouse CD45 (30-F11) (103107), APC-conjugated anti-mouse CD45.1 (A20) (110713), FITC-conjugated anti-mouse CD45.2 (104) (109805), APCCy7-conjugated anti-mouse CD45.2 (109824), PE-conjugated anti-mouse Gr1 (12-5931-82), and APC-conjugated anti-mouse Mac1 (M1/70) (17-0112-82) were from Biolegend or eBioscience. Annexin V (556419) and APC-conjugated anti-mouse CD8 (53–6.7) (553035) were from BD Pharmingen.

## Immunofluorescence staining

Paraffin embedded slides of tissue were dewaxed in three containers of fresh xylene for 5 min each, and then rehydrated in 100% ethanol twice for 10 min each, 95% ethanol twice for 10 min each, and dH$_2$O twice for 5 min each. Antigen retrieval was performed in pH 6.0 citrate buffer in water bath at 95℃ for 10 min and cooled slides on bench top for 30 min, and then rinsed in dH$_2$O for three times. Slides were then blocked in TBST (0.1% Tween-20) with 5% normal goat serum for 1 hr at room temperature and then incubated with primary antibody diluted in blocking buffer overnight at 4℃. Before staining with anti-pBAD(S136), primary antibody solution was incubated twice with membranes from SDS-PAGE loaded with *Bad*$^{-/-}$ mouse joint extracts overnight to eliminate nonspecific bindings and immunoblotting analysis was performed to confirm the elimination of nonspecific bindings. Next day, slides were washed with TBST three times for 5 min each and incubated with corresponding Alexa-Fluor 488 and Alexa-Fluor 594 labeled secondary antibodies (Invitrogen) diluted in blocking buffer for 2 hr at room temperature in dark. The slides were washed with TBST three times for 5 min each, incubated with DAPI (Sigma) and washed with TBST three times for 5 min each. Slides were mounted with fluorescent mounting medium (Dako) and visualized by Olympus BX51 or Nikon Eclipse Ti2 microscope. Validation of antibodies used for immunofluorescence staining was showed in *Supplementary file 1*.

## Immunohistochemistry staining

Paraffin embedded slides of mice or patients were dewaxed in xylene three times for 5 min each, and then rehydrated in 100% ethanol twice for 10 min each, 95% ethanol twice for 10 min each, and dH$_2$O twice for 5 min each. Antigen retrieval was performed in pH 6.0 citrate buffer in water bath at 95℃ for 10 min and cooled slides on bench top for 30 min and then rinsed slides in dH$_2$O for three times. Incubated slides with 3% hydrogen peroxide for 10 min and rinsed slides in dH$_2$O for two times. Slides were blocked in TBST with 5% normal goat serum for 1 hr at room temperature and then incubated with primary antibody diluted in blocking buffer overnight at 4℃. Slides were washed with TBST three times for 5 min each and incubated with one to three drops of SignalStain Boost Detection Reagent (#8114, CST) in a humidified chamber for 1 hr at room temperature. Slides were then washed with TBST three times for 5 min each and applied with 200 μl diluted SignalStain DAB (#8059, CST) for each section according to manufacturer instructions. Slides were immersed in dH$_2$O and counterstained with Hematoxylin (#14166, CST) for 1 min. Sections were washed in dH$_2$O twice for 5 min each and dehydrated in 95% ethanol twice for 10 s each, 100% ethanol twice for 10 s each, and xylene twice for 10 s each. Sections were then mounted with coverslips using mounting medium and visualized by Olympus BX51 or Nikon Eclipse Ti2 microscope.

## TUNEL assay

Apoptosis in paraffin embedded slides of synovium from CIA mice was detected using In Situ Cell Death Detection Kit (Cat No.11684817910, Roche) according to manufacturer's instructions. In brief, paraffin embedded slides were dewaxed in xylene three times for 5 min each, and then rehydrated in 100% ethanol twice for 10 min each, 95% ethanol twice for 10 min each, and dH$_2$O twice for 5 min each. Antigen retrieval was performed in 20 μg/ml Protein K solution for 20 min at 37℃ and

rinsed slides in PBS for five times. Then sections were incubated with TUNEL reaction mixture (prepared freshly before use) at 37°C for 1 hr in humidified chamber. Slides were incubated with DAPI (Sigma) and washed with PBS three times for 5 min each and mounted with fluorescent mounting medium (Dako) and visualized by Olympus BX51 or Nikon Eclipse Ti2 microscope.

## Immunoblotting

Total proteins of snap frozen joints were extracted in RIPA buffer (50 mM Tris-HCl, pH 7.6, 150 mM NaCl, 1% deoxycholic acid sodium salt, 0.1% SDS, 1% NP-40, 1 mM EDTA, and 1 mM EGTA) with phosphatase and protease inhibitors (1 mM DTT, 1 mM PMSF, 1 mM NaF, 1 mM $Na_3VO_4$, 1 μg/ml leupeptin, 1 μg/ml aprotinin, and 1 μg/ml pepstatin). Samples were rotated at 4°C for 30 min. The protein concentration was measured by BCA assay. Equal amounts of proteins were subjected to SDS-PAGE gels. Proteins were transferred to PVDF membranes (Millipore). Membranes were blocked in 5% (w/v) skim milk or 3% bovine serum albumin (BSA) diluted in TBST for 1 hr at room temperature and then incubated with diluted primary antibody overnight at 4°C. Next day, membranes were washed with TBST three times for 10 min each and then incubated with HRP-linked anti-rabbit IgG (#7074, CST) or HRP-linked anti-mouse IgG (#7076, CST) secondary antibody for 1 hr at room temperature. Membranes were washed with TBST three times for 10 min each and then exposed by enhanced chemiluminescence method. Original blots for immunoblotting analysis were showed in *Supplementary file 2*.

## Real-time quantitative PCR

Total RNA of snap frozen joints or cultured cells was extracted with TRIZOL reagent (No. 15596–018, Invitrogen), according to the manufacturer's instructions. Reverse transcription of RNA to complementary DNA was performed using M-MLV Reverse Transcriptase (M1705, Promega). Real-time quantitative PCR (qPCR) was performed using SYBR Premix Ex Taq (#RR420A, TaKaRa) and the program for qPCR was 95°C for 30 s, 40 cycles of 95°C for 10 s, 55°C for 30 s, and 72°C for 30 s. The following primers were used for mRNA expression detection: TNFα (forward: CCAAGGCGCCACATC TCCCT; reverse: GCTTTCTGTGCTCATGGTGT), IL-6 (forward: TAGTCCTTCCTACCCCAATTTCC; reverse: TTGGTCCTTAGCCACTCCTTC), IL-1β (forward: GAAGAAGAGCCCATCCTCTG; reverse: TCATCTCGGAGCCTGTAGTG), IL-17A (forward: TTTAACTCCCTTGGCGCAAAA; reverse: C TTTCCCTCCGCATTGACAC), IL-4 (forward: GGTCTCAACCCCCAGCTAGT; reverse: GCCGATGATC TCTCTCAAGTGAT), IFN-γ (forward: ATGAACGCTACACACTGCATC; reverse: CCATCC TTTTGCCAGTTCCTC), IL-21 (forward: GGACCCTTGTCTGTCTGGTAG; reverse: TGTGGAGCTGA TAGAAGTTCAGG), IL-10 (forward: GCTGGACAACATACTGCTAACC, reverse: ATTTCCGA TAAGGCTTGGCAA), MMP-3 (forward: ACATGGAGACTTTGTCCCTTTTG, reverse: TTGGCTGAG TGGTAGAGTCCC), MMP-13 (forward: CTTCTTCTTGTTGAGCTGGACTC, reverse: CTGTGGAGG TCACTGTAGACT), TRAP (forward: CACTCCCACCCTGAGATTTGT, reverse: CATCGTCTGCACGG TTCTG), Cstk (forward: GAAGAAGACTCACCAGAAGCAG, reverse: TCCAGGTTATGGGCAGAGA TT), and β-actin (forward: GGCTGTATTCCCCTCCATCG, reverse: CCAGTTGGTAACAATGCCATG T). Relative mRNA expression levels were calculated using the $2^{-\Delta\Delta Ct}$ method.

## ELISA

For titration of collagen II-specific antibodies in CIA, 96-well plate (#9018, Corning) was coated with 100 μl of 5 μg/ml collagen II overnight at 4°C, followed by blocking with PBS containing 1% (w/v) BSA for 1 hr at room temperature. Diluted serum (begin at 1:100) was applied and incubated at room temperature for 2 hr. HRP-conjugated goat anti-mouse IgG (1030–05), IgG1 (1070–05), IgG2b (1090–05), IgG2c (1708–05), and IgG3 (1100–05), all of which were from SouthernBiotech, were added and incubated at room temperature for 2 hr, followed by incubation with 50 μl/well TMB (P0209, Beyotime) for 20 min in dark at room temperature and stopped by 25 μl/well 1M $H_2SO_4$. Color development was monitored at 450/540 nm by a microplates reader.

For detection of cytokines in the serum of mice with CIA, ELISA kits for mouse IL-1β (MLB00C, R and D Systems), mouse IL-6 (M6000B, R and D Systems), and mouse TNFα (MTA00B, R and D Systems) were used, according to the manufacturer's instructions.

For detection of anti-dsDNA autoantibody in the serum of mice under normal condition, the Mouse anti-dsDNA IgG ELISA kit (5120, Alpha Diagnostic) was used, according to the manufacturer's instructions.

## Flow cytometry

For flow cytometry of infiltrated cells in the joints of mice with CIA, the hind paws were harvested. After the skins were removed, the paws were minced and digested in collagenase II (2 mg/ml; Sigma C6885) and DNase I (0.1 mg/ml; Roche, 776785) in 5 ml HBSS at 37°C for 2 hr with vortex every 30 min. The digested cells were washed and passed through 70 µm cell strainers (BD Biosciences). Cells were counted and stained with cell surface markers or Annexin V, or fixed, permeabilized, and stained with intracellular cell marker or cleaved Caspase-3, and finally analyzed using LSR II (BD Biosciences) or CytoFLEX (Beckman) flow cytometer.

For flow cytometry of cells from spleen and lymph nodes (LNs) of mice with CIA, the spleen and LNs were isolated and washed in PBS containing 1% (w/v) BSA and minced into single-cell suspension. Cells isolated from spleen were subjected to red blood cell lysis. Isolated cells were counted and incubated with fluorochrome-conjugated antibodies for 30 min on ice. Stained cells were analyzed using LSR II flow cytometer.

For flow cytometry of cells from peripheral blood, about 10 µl peripheral blood was taken from the mouse tail vein and mixed with 100 µl 10 mM EDTA, and centrifuged at $300 \times g$ for 5 min at 4°C. Supernatant was removed and added 1 ml ACK buffer to remove red blood cells. Cells were then centrifuged again and resuspended with 100 µl staining buffer (0.1% BSA in PBS) to do staining.

For flow cytometry of BMDMs, about 8-week-old mice were euthanized by $CO_2$ and BMDMs were isolated from femur and tibia bones of mice and cultured in L929 conditioned completed DMEM medium supplemented with 10% (v/v) FBS and 1% penicillin/streptomycin (100 U/ml) at 37°C with 5% $CO_2$ for 7 days after removing red blood cells. After treatment, cells were collected by TE without EDTA, counted, and washed with PBS, then centrifuged at $300 \times g$ for 5 min at 4°C and resuspended with 100 µl Annexin V staining buffer. Cells were added 5 µl Annexin V for each tube and incubated at room temperature for 15 min in dark. Then cells were added 400 µl Annexin V staining buffer and analyzed using LSR-Fortessa 4–15 HTS flow cytometer (BD Biosciences).

## In vitro osteoclastogenesis

Bone marrow cells were harvested from femur and tibia bones. After removing red blood cells by adding ACK lysis buffer, cells were resuspended in DMEM medium containing 10% FBS, 20% L929 medium, and 1% penicillin/streptomycin, and then plated in 12-well plates. Medium was changed on day 3. After 7 days, bone marrow macrophages were cultured in the presence of 50 ng/ml M-CSF (416 ML, R and D systems) and 100 ng/ml RANKL (315–11, PeproTech) or 50 ng/ml M-CSF alone for another 7 days. The culture medium was changed every other day. Cells were then fixed and stained with TRAP (leukocyte acid phosphatase kit #387A; Sigma-Aldrich) and TRAP-positive multinucleated cells (TRAP+ MNCs) were counted using light microscopy.

## TRAP staining

Paraffin embedded slides of joints were dewaxed in xylene and rehydrated in gradient ethanol and then incubated with the TRAP staining solution (leukocyte acid phosphatase kit #387A; Sigma-Aldrich), according to the manufacturer's instructions. TRAP+ MNCs were counted using light microscopy.

## Statistical analysis

Statistical analysis was performed using unpaired two-tailed Student's *t*-test for comparison of two groups and using one-way ANOVA test for comparison of four groups. The severity of clinical score of CIA model was evaluated by Mann–Whitney *U*-test. For all statistical analysis, $p < 0.05$ was considered statistically significant.

## Acknowledgements

This work was supported by National Institutes of Health grants (GM103868 to AL; CA195526 to JX; AI079087 and HL130724 to DW), and National Natural Science Foundation of China (31430026 and 91329301 to AL).

## Additional information

### Funding

| Funder | Grant reference number | Author |
| --- | --- | --- |
| National Institutes of Health | GM103868 | Anning Lin |
| National Institutes of Health | CA195526 | Jialing Xiang |
| National Institutes of Health | AI079087 | Demin Wang |
| National Institutes of Health | HL130724 | Demin Wang |
| National Natural Science Foundation of China | 31430026 | Anning Lin |
| National Natural Science Foundation of China | 91329301 | Anning Lin |

The funders had no role in study design, data collection and interpretation, or the decision to submit the work for publication.

### Author contributions

Jie Li, Data curation, Formal analysis, Validation, Investigation, Visualization, Methodology, Writing - original draft; Liansheng Zhang, Validation, Investigation, Methodology, Writing - original draft; Yongwei Zheng, Weida Yu, Investigation; Rui Shao, Methodology; Qianqian Liang, Resources; Hongyan Wang, Weiguo Zou, Writing - review and editing; Demin Wang, Resources, Funding acquisition, Writing - review and editing; Jialing Xiang, Resources, Funding acquisition; Anning Lin, Conceptualization, Supervision, Funding acquisition, Writing - original draft, Project administration, Writing - review and editing

### Author ORCIDs

Jie Li https://orcid.org/0000-0002-5446-894X
Liansheng Zhang https://orcid.org/0000-0001-6444-6421
Weiguo Zou http://orcid.org/0000-0003-2516-0302
Anning Lin https://orcid.org/0000-0003-2754-5134

### Ethics

Human subjects: Paraffin-embedded slides of synovial tissue from RA patients and OA patients were obtained from the Department of Pathology's diagnostic archive of the University of Chicago with diagnostic reports with patients' consent. This study was reviewed and approved by an Institutional Review Board (IRB) at the University of Chicago.

Animal experimentation: Animal studies were approved by the Institutional Animal Care and Use Committee of Shanghai Institute of Biochemistry and Cell Biology (Permit Number: SIBCB-S332-1607-016) or the Institutional Animal Care and Use Committee of the University of Chicago (#71840).

### Decision letter and Author response

Decision letter https://doi.org/10.7554/eLife.56309.sa1
Author response https://doi.org/10.7554/eLife.56309.sa2

## Additional files

### Supplementary files

• Supplementary file 1. Validation of antibodies used for immunofluorescence staining. (A) Immuno-blotting analysis of BAD. (B) Immunoblotting analysis of pBAD(S136) in joint extracts after removing nonspecific bands using $Bad^{-/-}$ joint extracts. (C) Immunoblotting analysis of Vimentin. (D) Immuno-blotting analysis of CD45R. (E) Immunoblotting analysis of F4/80. (F) Immunoblotting analysis of CD3. (G) Immunoblotting analysis of cleaved Casp-3 in mouse embryonic fibroblast (MEF) cells treated with TNFα (5 ng/ml) plus cycloheximide (CHX, 10 µg/ml). (H) Immunoblotting analysis of pBAD(S99) in THP-1 cells treated with TNFα (5 ng/ml). (I) Immunoblotting analysis of CD3 in THP-1 and Jurkat cell extracts. (J) Immunoblotting analysis of Vimentin in THP-1 and Hela cell extracts. (K) Immunoblotting analysis of CD68 in THP-1 and Jurkat cell extracts.

• Supplementary file 2. Original blots for immunoblotting analysis in *Figure 2—figure supplement 4A*, *Figure 4A*, *Figure 5E*, *Figure 7A*, and *Figure 7G*, as indicated.

• Transparent reporting form

### Data availability

All data generated or analysed during this study are included in the manuscript and supporting files. Source data files have been provided for all figures.

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
