## [Decision Letter]

**Acceptance summary:**

This study elegantly combines genetic and immunological approaches to reveal the role of the pro-apoptotic protein BAD in the inflammatory response associated with rheumatoid arthritis. The authors provide a link between inflammation and the inactivation of BAD as a key pathological mechanism leading to the survival of macrophages in the synovium. This discovery could pave the way for new therapeutic interventions centered on the modulation of BAD function in this disease.

**Decision letter after peer review:**

Thank you for submitting your article "BAD inactivation exacerbates rheumatoid arthritis pathology by promoting survival of sublining macrophages" for consideration by *eLife*. Your article has been reviewed by two peer reviewers, and the evaluation has been overseen by a Reviewing Editor and Satyajit Rath as the Senior Editor. The following individuals involved in review of your submission have agreed to reveal their identity: Jan P Tuckermann (Reviewer #1); Richard Pope (Reviewer #2).

The reviewers have discussed the reviews with one another and the Reviewing Editor has drafted this decision to help you prepare a revised submission.

Summary:

This is a carefully conducted study on the functional role of BAD in the pathogenesis of experimental arthritis and in rheumatoid arthritis (RA). The study makes use of independent models of experimental arthritis (collagen-induced arthritis and to a lesser extent TNF-tg induced arthritis) together with knockout mice. Disabling BAD function is shown to reduce apoptosis of macrophages in the sublinear layer and thereby lead to exaggerated inflammatory responses and subsequent erosion. The study also reports increased BAD phosphorylation in synovial tissue macrophages in RA compared with OA, supporting the clinical relevance of their finding in murine models. Overall, this is an important contribution to the field. However, the conclusion that the phenotype described is solely due to the loss of BAD function in macrophages is not entirely supported by the data.

The following comments are made to clarify the results presented and conclusions drawn.

Essential revisions:

1) The use of mice with conditionally deleted or mutated *Bad* would further strengthened the importance of BAD in macrophages in experimental arthritis. If such data is available, it should be included. While we acknowledge that this would be a major effort if the mice are not available to the authors, the authors should in such a case discuss that further studies would be needed to unambiguously ascribe a function role of BAD in macrophages and refer to the role of BAD in radio sensitive hematopoietic cells.

2) The authors use 2-color IHC to a large extent to identify cell types that are positive for TUNNEL, activated caspase 3, and pBAD. The approach would be strengthened if some of these observations were confirmed by flow cytometry. Figure 3F does not seem totally consistent with Figure 3G or Figure 5D with Figure 5B.

3) A clarification of the cell types positive for cleaved caspase 3 is merited. For example, why in the entire ankle cleaved caspase 3 activity is reduced in BAD deficient animals, whereas on histology only macrophages seemed to be affected (which make up to 10% of the tissue cells?).

4) Important controls should be included. For example, the levels of pBAD in non-arthritic joints (e.g. in Figure 4D, E) should be shown. The basal expression of BAD in the distinct cell populations (FLS, B cells, T cells, Neutrophils) should be elaborated and compared to macrophages.

5) Please be consistent when referring to CIA as an experimental model of RA. In some places, experimental arthritis is incorrectly called RA.

---

## [Author Response]

Essential revisions:1) The use of mice with conditionally deleted or mutated Bad would further strengthened the importance of BAD in macrophages in experimental arthritis. If such data is available, it should be included. While we acknowledge that this would be a major effort if the mice are not available to the authors, the authors should in such a case discuss that further studies would be needed to unambiguously ascribe a function role of BAD in macrophages and refer to the role of BAD in radio sensitive hematopoietic cells.

We totally agree with the reviewers that the usage of mice with conditionally deleted or mutated *Bad* would further strengthen the importance of BAD in macrophages in experimental arthritis, given the possibility that *Bad* loss may affect the radio sensitive hematopoietic cells. Unfortunately, these mouse strains are not available for us. In the previous manuscript, we used bone marrow transplantation model to demonstrate the important role of BAD in macrophages (Figure 3). In the revised manuscript, we further analyzed the sensitivity of isolated synovial macrophages, B cells, CD4^+^ T cells and fibroblasts from WT and *Bad^-/-^* mice in CIA model to apoptosis by flow cytometry (new Figure 2C, D, new Figure 2—figure supplement 5, new Figure 2—figure supplement 6, new Figure 2—figure supplement 7). The new data are consistent with our previous conclusion that *Bad*-deficient macrophages, but not B cells, CD4^+^ T cells or fibroblasts, are resistant to apoptosis in vivo. In the revised Discussion, we have indicated that such conditional *Bad* knockout or mutant mice are needed to unambiguously ascribe the role of BAD in macrophages, given the possibility that *Bad* loss may affect the radio sensitive hematopoietic cells.

2) The authors use 2-color IHC to a large extent to identify cell types that are positive for TUNNEL, activated caspase 3, and pBAD. The approach would be strengthened if some of these observations were confirmed by flow cytometry. Figure 3F does not seem totally consistent with Figure 3G or Figure 5D with Figure 5B.

We thank the reviewers for these great suggestions. As suggested by the reviewers, in the revised manuscript we performed flow cytometry analysis with Alexa Fluor 488-conjugated cleaved caspase-3 antibody to compare the ratio of apoptotic cells among macrophages, B cells, CD4^+^ T cells and synovial fibroblasts. The results show that the ratio of apoptotic synovial macrophages, but not other cell types examined, was significantly reduced both in *Bad^-/-^* mice in CIA model and TNF-Tg/*Bad^-/-^* mice (new Figure 2C, D, new Figure 2—figure supplement 6, and new Figure 6F, G), consistent with our previous data showing that macrophage is the major cell type significantly affected by *Bad* loss using TUNEL assay, activated caspase-3 and pBAD staining (Figure 2—figure supplement 4, Figure 2A, B and Figure 4D, E). The same results were obtained when Annexin V was used as another apoptotic indicator to label apoptotic cells isolated from the synovium (new Figure 2—figure supplement 5 and new Figure 2—figure supplement 7). These data further support our conclusion that *Bad* loss mainly affects the apoptosis of synovial macrophages in experimental arthritic murine models used. In addition, we replaced the pictures in Figure 3F and Figure 5B to make the figures to be consistent with the statistical analysis.

3) A clarification of the cell types positive for cleaved caspase 3 is merited. For example, why in the entire ankle cleaved caspase 3 activity is reduced in BAD deficient animals, whereas on histology only macrophages seemed to be affected (which make up to 10% of the tissue cells?).

We thank the reviewers for the excellent questions. The reviewer is correct that the cleaved caspase-3 activity is reduced in the joint extracts of *Bad^-/-^* mice compared with WT mice in CIA model, indicating that *Bad* loss inhibits the apoptosis of the cells in experimental arthritis. To identify which cell type is mainly affected by *Bad* loss, we did TUNEL staining along with different cell markers and found that macrophages were mainly affected among different cell types in the joint (revised Figure 2—figure supplement 4). In addition, cleaved caspase-3 staining assay with macrophage marker further demonstrated that there was a difference in macrophage apoptosis between WT and *Bad^-/-^* mice joints in CIA model (Figure 2A, B). Furthermore, cleaved caspase-3 flow cytometry assay demonstrated that *Bad* loss mainly affected macrophage apoptosis (new Figure 2C, D, new Figure 2—figure supplement 6). The reason why only macrophage apoptosis is mainly affected is most likely due to distinct expression levels of BAD in different types of infiltrating immune cells, as discussed in the revised manuscript Discussion. Thus, synovial sublining macrophages may be more dependent on inactivation of BAD to resist apoptosis than other immune cells in the arthritic joints. To avoid any confusion, in the revised manuscript we stated that *Bad* loss mainly affects macrophage apoptosis, while apoptosis of other cell types appear to be less affected.

4) Important controls should be included. For example, the levels of pBAD in non-arthritic joints (e.g. in Figure 4D, E) should be shown. The basal expression of BAD in the distinct cell populations (FLS, B cells, T cells, Neutrophils) should be elaborated and compared to macrophages.

We thank the reviewers for these great suggestions. Accordingly, we have included the controls of BAD staining in arthritic joints (new Figure 4—figure supplement 1) and pBAD staining in non-arthritic joints (new Figure 4—figure supplement 2) for Figure 4D, E. We also included the data of BAD staining in RA synovium (new Figure 8—figure supplement 1) as the control of Figure 8C, D in the revised manuscript.

5) Please be consistent when referring to CIA as an experimental model of RA. In some places, experimental arthritis is incorrectly called RA.

We apology for the ignorance and have corrected this mistake.